# The Shc1 adaptor simultaneously balances Stat1 and Stat3 activity to promote breast cancer immune suppression

Ryuhjin Ahn[1,2], Valérie Sabourin[1], Alicia M. Bolt[1,3], Steven Hébert[1], Stephanie Totten[1,2], Nicolas De Jay[1,4], Maria Carolina Festa[1], Yoon Kow Young[1], Young Kyuen Im[1,2], Tony Pawson[5], Antonis E. Koromilas[1,2,3], William J. Muller[1,2,6,7], Koren K. Mann[1,2,3], Claudia L. Kleinman[1,4] & Josie Ursini-Siegel[1,2,3,6,7]

Tyrosine kinase signalling within cancer cells is central to the establishment of an immuno-suppressive microenvironment. Although tyrosine kinase inhibitors act, in part, to augment adaptive immunity, the increased heterogeneity and functional redundancy of the tyrosine kinome is a hurdle to achieving durable responses to immunotherapies. We previously identified the Shc1 (ShcA) scaffold, a central regulator of tyrosine kinase signalling, as essential for promoting breast cancer immune suppression. Herein we show that the ShcA pathway simultaneously activates STAT3 immunosuppressive signals and impairs STAT1-driven immune surveillance in breast cancer cells. Impaired Y239/Y240-ShcA phosphor-ylation selectively reduces STAT3 activation in breast tumours, profoundly sensitizing them to immune checkpoint inhibitors and tumour vaccines. Finally, the ability of diminished tyrosine kinase signalling to initiate STAT1-driven immune surveillance can be overcome by compensatory STAT3 hyperactivation in breast tumours. Our data indicate that inhibition of pY239/240-ShcA-dependent STAT3 signalling may represent an attractive therapeutic strategy to sensitize breast tumours to multiple immunotherapies.

[1] Lady Davis Institute for Medical Research, 3755 Chemin de la Côte-Sainte-Catherine, Montréal, Quebec, Canada H3T 1E2. [2] Department of Experimental Medicine, McGill University, 845 Rue Sherbrooke O, Montréal, Quebec, Canada H3A 0G4. [3] Department of Oncology, McGill University, 546 Pine Avenue West, Montréal, Quebec, Canada H2W 1S6. [4] Department of Human Genetics, McGill University, 1205 Dr. Penfield Ave, Montréal, Quebec, Canada H3A 1B1. [5] The Lunenfeld-Tanenbaum Research Institute, Sinai Health System, 600 University Ave., Toronto, Ontario, Canada M5G 1X5. [6] Department of Biochemistry, McGill University, McIntyre Medical Centre, Montréal, Quebec, Canada H3G 1Y6. [7] Goodman Cancer Research Centre, 1160 Avenue des Pins, Montréal, Quebec, Canada H3A 1A3. Correspondence and requests for materials should be addressed to J.U.-S. (email: giuseppina.ursini-siegel@mcgill.ca).

Immunotherapy, which attempts to bolster the patient's own immune system, represents an intense area of cancer research. Diverse immunotherapies are in clinical trials including the following: (1) vaccines, which stimulate immune responses against tumour antigens; (2) monoclonal antibodies, which promote immune-mediated cytotoxicity; and (3) oncolytic viruses and (4) immune checkpoint inhibitors, which overcome T-cell anergy[1]. These therapeutic approaches have significantly improved patient outcome in metastatic melanoma and non-small cell lung cancer[2,3]. Although immunotherapy for poor outcome breast cancers is in its infancy, pre-clinical studies support this approach. High numbers of tumour-infiltrating lymphocytes in HER2 and basal breast cancers serve as an independent predictor of good outcome[4–6]. Moreover, part of the therapeutic efficacy of Trastuzumab, a HER2-neutralizing antibody, relies on its ability to augment innate and adaptive immunity in breast cancer[7]. The induction of adaptive immunity also increases the anti-tumorigenic potential of anthracycline-based chemotherapies in estrogen receptor (ER)-negative breast cancers[8,9].

Recent studies have examined whether combining immunotherapies with targeted agents or chemotherapies prolonged survival in cancer patients[10]. Combining Trastuzumab with tumour vaccines led to a detectable, albeit modest, increase in disease-free survival in women with metastatic HER2+ breast cancer[11]. Thus, more effective strategies are required to improve these combination therapies. Numerous studies suggest that tyrosine kinases potentiate immune suppression. Epidermal growth factor receptor (EGFR) signalling in lung cancer activates the PD1 immune checkpoint to promote immune evasion[12] and an EGFR-neutralizing antibody stimulates immunogenic cell death in colorectal cancers[13]. Abrogating signalling downstream of the Ron or TAM family of receptor tyrosine kinases (RTKs) impaired the development of breast cancer lung metastases through re-activation of anti-tumour immune responses[14,15]. Finally, the FAK tyrosine kinase regulates transcriptional responses that block anti-tumour immunity[16]. An important caveat that may limit the efficacy of tyrosine kinase inhibitors in augmenting tumoricidal immune responses is the inherent functional redundancy within the tyrosine kinome, leading to the emergence of therapeutic resistance[17].

Tyrosine kinases rely on a core set of signalling intermediates to transduce oncogenic signals. One such scaffolding protein, called Shc1 (or ShcA), is recruited to multiple tyrosine kinases and is essential for tumour initiation, progression and metastatic spread in breast cancer mouse models[6,18,19]. The mammalian ShcA gene encodes three proteins that are generated through differential promoter usage (p66) or alternative translational start sites (p46 and p52)[20,21]. p46/52ShcA employs numerous domains and motifs to transduce phosphotyrosine-dependent signals downstream of tyrosine kinases[21–25]. ShcA translocates from the cytosol to the plasma membrane where it interacts with phosphotyrosine residues in activated tyrosine kinases. These interactions are mediated by either the PTB or SH2 domains of ShcA[23,26,27]. In turn, tyrosine kinases phosphorylate three tyrosine residues (Y239/Y240 and Y317 − analogous to Y313 in mice) within the central collagen homology 1 domain of ShcA[19,25,28]. Once phosphorylated, these tyrosines serve as docking sites for other PTB- and SH2-containing proteins to activate diverse pathways, including but not limited to the Ras/mitogen-activated protein kinase and phosphoinositide 3-kinase/AKT pathways[20,22]. We previously showed that tyrosine kinases require downstream ShcA signalling to evade anti-tumour immunity[6]. Herein we elucidate the mechanisms through which ShcA transduces immunosuppressive signals. We now show that the ShcA phosphotyrosine motifs potentiate immune suppression

by limiting signal transducer and activator of transcription (STAT)-1-driven anti-tumour immunity, while simultaneously increasing STAT3 immunosuppressive signals. We further demonstrate that attenuating ShcA signalling downstream of activated tyrosine kinases sensitizes mammary tumours to several immunotherapies.

## Results

**pY239/240 ShcA signalling contributes to immune suppression.** We previously established that ShcA-coupled tyrosine kinase signalling promotes breast cancer immune suppression[6]. Herein we employ 'knock-in' mice expressing ShcA mutant alleles that are debilitated in phosphotyrosine (pY)-dependent 239/240 (Y239/240F—2F) or 313 (Y313F—313F) signalling (Fig. 1a) under the control of the endogenous ShcA promoter[29]. Using MMTV/PyVmT (MT) transgenic mice, we previously showed that both ShcA phosphorylation sites are required for breast cancer development[19]. We chose the MMTV/MT mouse model for two reasons. First, MT-induced transformation recapitulates all the stages of breast cancer progression in transgenic mice[30]. Second, ShcA-coupled signal transduction is important for MT-induced breast cancer development[31]. We now show that loss of cytotoxic T lymphocyte-driven anti-tumour immunity had no effect on tumour onset in MT/ShcA$^{+/+}$ mice, consistent with published studies[32], and minimally accelerated tumour onset in MT/Shc$^{313F/313F}$ mice. In contrast, tumour onset was significantly accelerated in MT/Shc$^{2F/2F}$ females in a CD8$^{-/-}$ background ($P \leq 0.001$ by multiple $t$-test) (Fig. 1b). These data suggest that the Y239/240-ShcA phosphorylation sites transduce immunosuppressive signals. We next established cell lines from four to five independent MT/ShcA$^{+/+}$, MT/Shc$^{2F/2F}$ and MT/Shc$^{313F/313F}$ mammary tumours to test whether ShcA signalling in the epithelial compartment contributes to immune suppression. Tumour onset of two independent MT/ShcA$^{+/+}$, MT/Shc$^{2F/2F}$ and MT/Shc$^{313F/313F}$ cell lines was unaffected in CD8$^{+/+}$ versus CD8$^{-/-}$ (Supplementary Fig. 1a,b) or IFN$\gamma^{+/+}$ versus IFN$\gamma^{-/-}$ animals (Supplementary Fig. 1c,d). In contrast, the growth rate of two independent MT/Shc$^{2F/2F}$ breast tumours was selectively impaired in mice that contain CD8+ T cells (CD8$^{+/+}$) or retain intact interferon-$\gamma$ (IFN$\gamma$) responses (IFN$\gamma^{+/+}$) relative to their immune-deficient (CD8$^{-/-}$ and IFN$\gamma^{-/-}$) counterparts. In contrast, the presence of an intact immune response marginally reduced or unaffected the growth rate of two independent MT/ShcA$^{+/+}$ and MT/Shc$^{313F/313F}$ mammary tumours, suggesting that they are immunosuppressed (Fig. 1c,d and Supplementary Fig. 1e,f). Infiltration of CD3$^+$ T cells and Granzyme B$^+$ effector cells was significantly increased in MT/Shc$^{2F/2F}$ tumours ($P < 0.0001$ by Mann–Whitney $U$-test) (Fig. 1e and Supplementary Fig. 2a–c). Finally, CD8$^+$ T cells, including activated cytotoxic T lymphocytes (CD8$^+$CD69$^+$), are specifically recruited into MT/ShcA$^{2F/2F}$ tumours (Fig. 1f,g and Supplementary Fig. 2d). Furthermore, we observed a co-incident increase in recruitment of CD11b$^+$Gr1$^+$ myeloid-derived suppressor cells into the tumours and spleens of MT/Shc$^{2F/2F}$ mice, which may reflect compensatory immunosuppressive responses (Fig. 1f,g and Supplementary Fig. 2e). Moreover, differences in MT/ShcA$^{+/+}$, MT/Shc$^{2F/2F}$ and MT/Shc$^{313F/313F}$ cell morphology in vitro or tumour histology in vivo do not stratify whether breast tumours exhibit immune surveillance or immune suppression phenotypes (Supplementary Fig. 3). Combined, these data suggest that impaired Y239/240-ShcA phosphorylation specifically sensitizes mammary tumours to immune surveillance.

**Loss of pY313-ShcA signalling induces IFN$\gamma$ responses.** To define how tyrosine kinases engage ShcA to promote immune

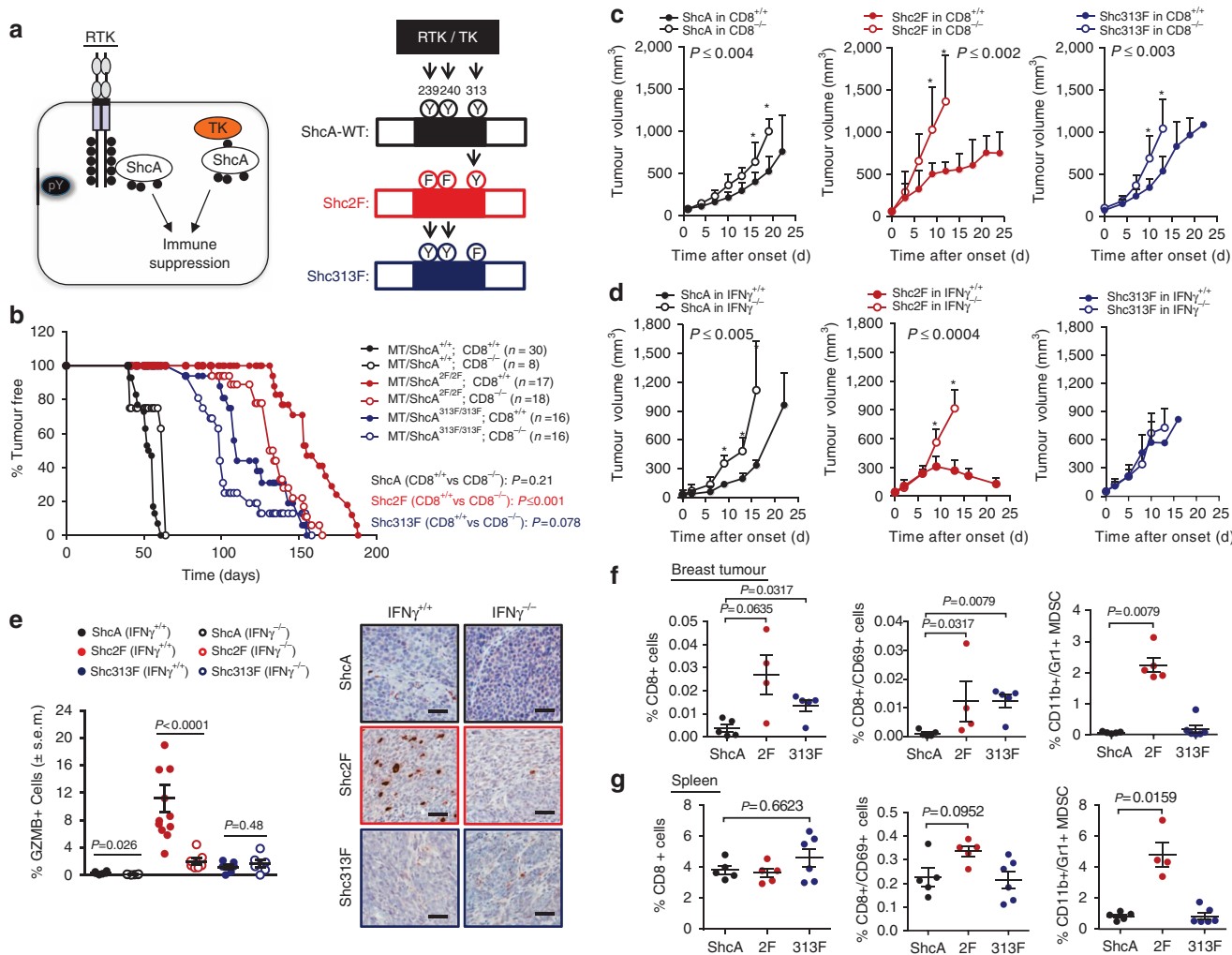

**Figure 1 | Phosphotyrosine-dependent ShcA signalling promotes breast cancer immune suppression.** (**a**) Schematic diagram illustrating engagement of TK/ShcA signalling complexes to promote immune suppression. (**b**) MMTV/MT transgenic mice of the indicated genotypes were evaluated for mammary tumour onset. Percentage of tumour-free mice over time. Number (n) of mice analysed is indicated. (**c,d**) Cell lines derived from MT-driven transgenic mammary tumours that are homozygous for WT *ShcA* (864) or phosphotyrosine-deficient *ShcA* mutants (Y313F (313F-6738)) or Y239F/Y240F (2F-5372) were injected into the fourth mammary fat pad of FVB, $CD8^{-/-}$ or $IFN\gamma^{-/-}$ mice. Tumour outgrowth was measured and represented as mean tumour volume ($mm^3$) ± s.e.m. (n = 8-12). (**e**) Immunohistochemical staining of tumour tissue (n = 6-12 per group) harvested from the indicated mice using Granzyme B (GZMB)-specific antibodies. The data are represented as percentage $GZMB^+$ cells relative to total cells per field ± s.e.m. Representative images are shown. Scale bars, 50 μm. (**f,g**) Flow cytometric analysis of immune infiltrates into $MT/ShcA^{+/+}$ (864), $MT/Shc^{2F/2F}$ (5372) and $MT/Shc^{313F/313F}$ (6738) (**f**) tumour tissue or (**g**) matching spleen derived from FVB mice. Presence of $CD8^+$ cells, $CD8^+CD69^+$ cells and $CD11b^+ Gr1^+$ MDSCs (n = 4-6 mice per group). The data are represented as percentage of each cell type relative to total cells analysed ± s.e.m. Significance was determined by Wilcoxon's rank-sum test for **e–g**, by multiple t-test with Holm–Sidak method for **c,d** (*statistically significant time points as indicated in the top left corner), and by two-tailed two sample t-test for **a,b**.

suppression, we performed RNA sequencing (RNAseq) on four independent $MT/ShcA^{+/+}$, $MT/Shc^{2F/2F}$ and $MT/Shc^{313F/313F}$ cell lines. Loss of distinct ShcA signals does not lead to global transcriptional changes (Supplementary Fig. 4), consistent with the fact that this signalling pathway regulates expression of a discrete set of genes. Instead, we identified 174 assigned genes that are differentially regulated by alterations in pY-driven ShcA signalling. These include transcriptional targets that are uniquely differentially expressed by loss of pY313 (98 genes) or pY239/240 (64 genes) ShcA signals (Fig. 2a and Supplementary Data 1). Surprisingly, we observed many IFN-responsive genes (42%) involved in host defense or immunity, specifically in tumours that are impaired in pY313-ShcA signalling (Fig. 2b). We validated that $MT/Shc^{313F/313F}$ breast cancer cells uniformly and basally upregulated many IFNγ-responsive genes, including CXCL9 and

components of the antigen processing and presentation (APP) machinery (Fig. 2c–e and Supplementary Fig. 5a–c) Finally, $MT/Shc^{313F/313F}$ breast cancer cells basally upregulated surface major histocompatibility complex (MHC) class I expression (Fig. 2f,g). We extended these observations to the MMTV/NIC transgenic mouse model, which expresses an oncogenic ErbB2 variant in the mammary epithelium[19]. Bigenic NIC/ShcA^{fl/fl} mammary tumours, which lack ShcA in the epithelial compartment, also upregulate several components of the APP machinery (Supplementary Fig. 5d).

These data suggest that although reduced pY313-ShcA signalling potentiates IFN-regulated transcriptional responses in breast cancer cells *in vitro*, it is insufficient to overcome immune suppression in $MT/Shc^{313F/313F}$ tumours *in vivo* (Fig. 1c,d). In contrast, inhibition of pY239/240ShcA-coupled signalling

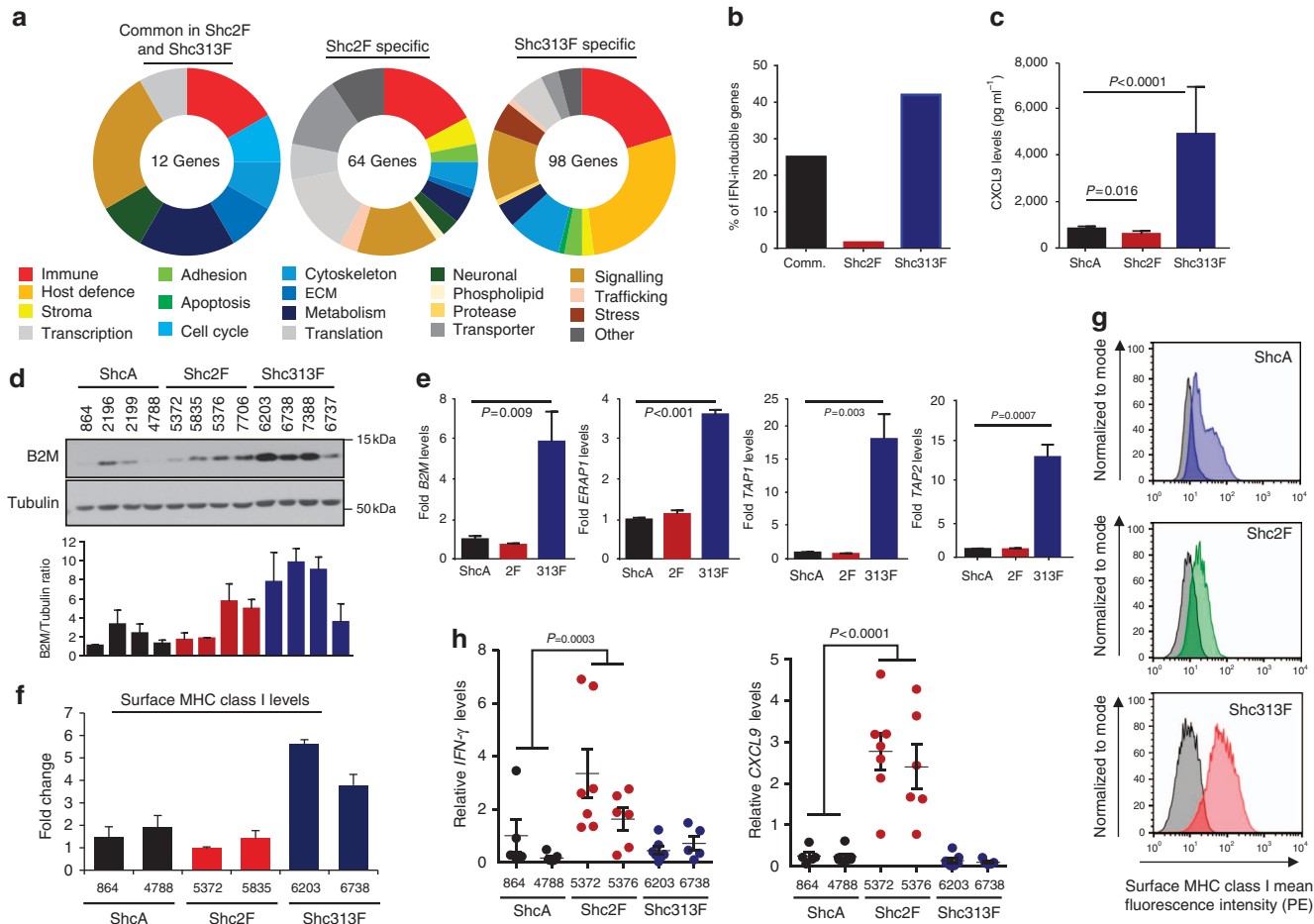

**Figure 2 | An intact ShcA pathway restrains IFN-driven immune responses in breast cancer cells.** (**a**) RNAseq was performed on cell lines derived from four independent MT/ShcA$^{+/+}$, MT/Shc$^{2F/2F}$ and MT/Shc$^{313F/313F}$ mammary tumours. Genes that were significantly differentially regulated in all four cell lines were identified and stratified into three groups: 2F-specific (64 genes), 313F-specific (98 genes) or commonly differentially expressed in 2F and 313F cells relative to WT ShcA (12 genes). (**b**) Percentage of IFN-regulated genes identified by RNAseq (**a**) within each signature. (**c**) Average CXCL9 protein levels (ng ml$^{-1}$) secreted from two independent MT/ShcA$^{+/+}$, MT/Shc$^{2F/2F}$ and MT/Shc$^{313F/313F}$ breast cancer cells ($n=5$–6 supernatants per cell line) following a 24 h IFNγ (1 ng ml$^{-1}$) stimulation as determined by ELISA ( ± s.d.). (**d**) Top: immunoblot analysis of the indicated breast cancer cell lines using B2M and Tubulin-specific antibodies. Bottom: average B2M levels, normalized to Tubulin, ± s.d., as quantified by Image J software ($n=3$ independent experiments). (**e**) Relative *B2m*, *ERAP1*, *TAP1* and *TAP2* mRNA levels (normalized to *GAPDH*) under basal conditions in MT/ShcA$^{+/+}$ (864), MT/Shc$^{2F/2F}$ (5372) and MT/Shc$^{313F/313F}$ (6738) breast cancer cells. The data are shown as average fold change relative to MT/ShcA$^{+/+}$ cells ± s.d. ($n=12$ per condition from three independent experiments). (**f**) Surface MHC class I levels of independent MT/ShcA$^{+/+}$, MT/Shc$^{2F/2F}$ and MT/Shc$^{313F/313F}$ breast cancer cell lines as determined by flow cytometry. The data are shown as average fold change relative to ShcA$^{+/+}$ (864) ± s.d. Representative of two independent experiments. (**g**) Representative histograms for surface MHC class I expression levels. Unstained control is in grey. (**h**) Independent MT/ShcA$^{+/+}$, MT/Shc$^{2F/2F}$ and MT/Shc$^{313F/313F}$ breast cancer cell lines were injected into the mammary fat pads of FVB mice. Tumours were analysed for relative *IFN-γ* and *CXCL9* levels (normalized to *GAPDH*) ± s.e.m. ($n=7$ tumours per group). To determine significance, Mann–Whitney *U*-test was performed to compare MT/ShcA$^{+/+}$ group (864 and 4788) with MT/Shc$^{2F/2F}$ (5372 and 5376) for **h** and two-sample *t*-test was performed for **c**,**e**.

pathways in mammary tumours (MT/Shc$^{2F/2F}$) specifically elicits immune surveillance *in vivo* (Fig. 1c,d and Supplementary Fig. 1), despite the fact that IFNγ-driven signalling responses are basally reduced in these cell lines *in vitro* (Fig. 2). This is consistent with the fact that *IFNγ* and *CXCL9* messenger RNA levels are specifically increased in MT/Shc$^{2F/2F}$ tumours but not in MT/ShcA$^{+/+}$ or MT/Shc$^{313F/313F}$ tumours (Fig. 2h). Thus, although loss of pY313-ShcA signalling in breast cancer cells basally upregulates IFNγ-inducible genes associated with anti-tumour immunity, they are restrained in MT/Shc$^{313F/313F}$ mammary tumours *in vivo*. These data suggest that additional immunosuppressive mechanisms are engaged in MT/Shc$^{313F/313F}$ tumours to restrain induction of IFNγ-driven anti-tumour immunity. In contrast, MT/Shc$^{2F/2F}$ tumours are sensitized to IFNγ-driven immune surveillance *in vitro*, even though

inhibition of pY239/240ShcA signalling does not upregulate IFNγ-stimulated activity *in vitro*. These data suggest that loss of pY239/240ShcA signalling downstream of tyrosine kinases inhibits immunosuppressive signals in breast cancer cells, which sensitizes them to stromally derived, IFNγ-inducible immune surveillance pathways *in vivo*.

**ShcA pathway potentiates STAT3 and represses STAT1 activity.** To better understand these seemingly paradoxical observations and with the knowledge that IFN signalling requires the STAT1 transcription factor, we examined whether ShcA signalling dynamically regulated the activity of STAT1 and STAT3, two transcription factors with opposing roles in immune evasion[33]. As expected, STAT1 levels are basally elevated in all

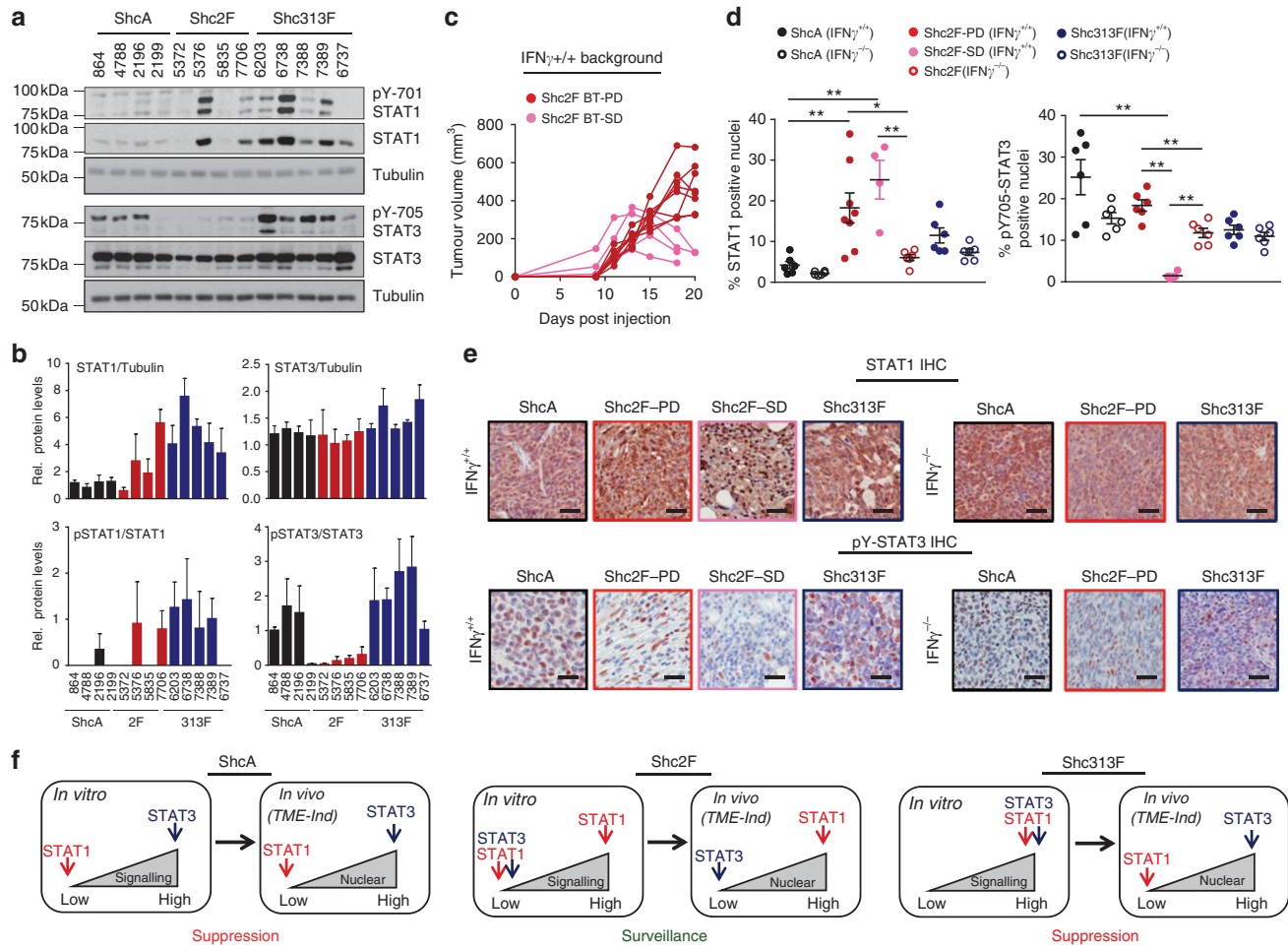

**Figure 3 | Distinct ShcA-driven phosphotyrosine signalling networks differentially activate STAT3 and STAT1 signalling in breast cancer cells.**
(**a**) Immunoblot analysis of total cell lysates from MT/ShcA$^{+/+}$, MT/Shc$^{2F/2F}$ and MT/Shc$^{313F/313F}$ breast cancer cells (four to five per genotype) using STAT1, pY701-STAT1, STAT3, pY705-STAT3 and Tubulin antibodies. (**b**) Densitometric quantification of immunoblots using ImageJ software. The data show average fold change in expression levels (as indicated) ± s.d. in the individual cell lines from three independent experiments. (**c**) Growth curves for individual tumour MT/Shc$^{2F/2F}$ (5372) mammary tumours that emerge in an immunocompetent FVB background (IFNγ$^{+/+}$). Each line describes the tumour volume (mm$^3$) of an individual breast tumour (BT) at the indicated days post injection and is representative of two independent experiments. PD, progressive disease (dark red dot), SD, stable disease (pink dot). (**d**) Immunohistochemical staining of mammary tumours that emerged in IFNγ$^{+/+}$ or IFNγ$^{-/-}$ mice using STAT1- and pY705-STAT3-specific antibodies ($n = 6$–8 tumours per genotype). The mean percentage of STAT1$^+$ and pY705-STAT3$^+$ stained nuclei ± s.e.m. is shown. MT/Shc$^{2F/2F}$ tumours that displayed PD or SD phenotypes were stratified. The data are representative of two independent experiments and significance was analysed by Wilcoxon's rank-sum test (*$P < 0.05$ and **$P < 0.01$). (**e**) Representative images of STAT1- and pY705-STAT3-stained paraffin-embedded sections. Scale bars, 50 μm. (**f**) Schematic diagram summarizing how altered pY239/240- and pY313-ShcA signalling affect STAT1 and STAT3 activation, both in established cell lines *in vitro* and in mammary tumours *in vivo* (induced by the tumour microenvironment, TME). The consequence of the individual ShcA tyrosine phosphorylation sites on emergence of pro- or anti-tumorigenic immune responses is also shown.

MT/Shc$^{313F/313F}$ cells, coinciding with increased Y701-STAT1 phosphorylation in most of them. In contrast, STAT1 levels are elevated in 50% of the MT/Shc$^{2F/2F}$ cell lines (Fig. 3a,b). However, MT/Shc$^{2F/2F}$ tumours that express low (5372) or high (5376) STAT1 levels are similarly susceptible to immune surveillance (Fig. 1c,d and Supplementary Fig. 1e,f). We extended our analysis to include the STAT3 transcription factor, which is essential for breast tumours to evade anti-tumour immunity[34]. STAT3 Y705 phosphorylation is uniformly reduced in MT/Shc$^{2F/2F}$ breast cancer cells (Fig. 3a,b). Taken together, these data suggest that tyrosine kinases engage the Y239/240-ShcA phosphorylation sites primarily to potentiate STAT3 immunosuppressive signals and use the Y313-ShcA phosphorylation sites to attenuate STAT1-dependent anti-tumour immunity. Moreover, IFNα, IFNβ and IFNγ expression levels are absent or exceedingly low in MT/Shc$^{313F/313F}$ cells

(Supplementary Table 1), suggesting that the increased STAT1 activity observed in response to impaired pY313-ShcA signalling does not result from the establishment of an IFN-dependent autocrine loop.

Analysis of MT/Shc$^{2F/2F}$ tumours identified two populations that proceed through distinct growth phases, specifically in immunocompetent mice. One subset of tumours displayed progressive growth (PD: ~60%), whereas another proceeded through a plateau growth phase and was more reminiscent of stable disease (SD: ~40%) (Fig. 3c). Moreover, this plateau phase is immune-mediated given that all MT/Shc$^{2F/2F}$ tumours display progressive growth in immune-deficient mice (Fig. 3c). Given these different growth patterns, we also examined how perturbation in phosphotyrosine-dependent ShcA signalling impacts the STAT1 and STAT3 activation status of mammary tumours that display immune surveillance (MT/

$Shc^{2F/2F}$) versus suppression (MT/ShcA$^{+/+}$ and MT/Shc$^{313F/313F}$) phenotypes. All MT/Shc$^{2F/2F}$ tumours displayed an IFN$\gamma$-inducible increase in nuclear STAT1 levels relative to MT/ShcA$^{+/+}$ and MT/Shc$^{313F/313F}$ tumours, which are immuno-suppressive. In contrast, the percentage of nuclear pY705-STAT3$^{+}$ cells are elevated in MT/ShcA$^{+/+}$ and MT/Shc$^{313F/313F}$ tumours (Fig. 3d,e and Supplementary Fig. 6). Although MT/Shc$^{2F/2F}$ breast cancer cells display reduced Y705-STAT3 phosphorylation in vitro (Fig. 3a), progressively growing MT/Shc$^{2F/2F}$ tumours (PD) that evolved in an immunocompetent background hyperactivated STAT3 compared with those that emerged in IFN$\gamma^{-/-}$ mice. Thus, re-acquisition of STAT3 activity in MT/Shc$^{2F/2F}$ tumours is associated with the eventual emergence of progressively growing tumours. In contrast, pY705-STAT3 nuclear staining is virtually ablated in MT/Shc$^{2F/2F}$ tumours that resemble SD (Fig. 3d,e). These data suggest that debilitated pY239/240-ShcA signalling reduces STAT3 activation in breast tumours to sensitize them to immune surveillance. They further suggest that increased STAT1 signalling in MT/Shc$^{313F/313F}$ mammary tumours is insufficient to promote immune suppression, because they retain elevated STAT3 activity.

**STAT3 activity dictates STAT1-driven anti-tumour immunity.** To interrogate the functional significance of STAT1 in limiting the tumorigenic potential of breast tumours with low (MT/Shc$^{2F/2F}$) versus high (MT/ShcA$^{+/+}$; MT/Shc$^{313F/313F}$) STAT3 activity, we deleted STAT1 by CRISPR/Cas9 genomic editing (Fig. 4a and Supplementary Fig. 7a). Although STAT1 loss increased Y705-STAT3 phosphorylation in IFN$\gamma$-stimulated MT/ShcA$^{+/+}$ cells, it did not rescue pY705-STAT3 levels in MT/Shc$^{2F/2F}$ cells, suggesting that the Y239/240-ShcA phosphorylation sites directly regulate Y705-STAT3 phosphorylation. Moreover, pY705-STAT3 levels remained constant in MT/Shc$^{313F/313F}$ cells, suggesting that increased STAT1 activity in these cells promoted a compensatory activation of STAT3 signalling to maintain immune suppression (Fig. 4a). We also deleted STAT3 from MT/ShcA$^{+/+}$ and MT/Shc$^{313F/313F}$ cells (Fig. 4b and Supplementary Fig. 7b). STAT3 loss had no effect on baseline STAT1 expression levels, suggesting that STAT3 does not reciprocally regulate STAT1. We next assessed surface MHC class I levels in MT/ShcA$^{+/+}$, MT/Shc$^{2F/2F}$ and MT/Shc$^{313F/313F}$ cell lines (control, STAT1 deficient or STAT3 deficient) via flow cytometry. STAT1, but not STAT3, increased basal (MT/Shc$^{313F/313F}$) and IFN$\gamma$-inducible (MT/ShcA$^{+/+}$; MT/Shc$^{2F/2F}$) surface MHC class I expression on breast cancer cells (Fig. 4c and Supplementary Fig. 7c). We also examined expression levels of several IFN-stimulated genes (TAP2, IRF9 and DDX60) and show that STAT1 deficiency abrogates their basal and IFN$\gamma$-inducible expression levels in MT/Shc$^{313F/313F}$ cells (Fig. 4d). These data further support the observation that pY313-ShcA signalling restrains STAT1-mediated transcriptional responses. Published studies show that MUC1 is a STAT3 target gene[35]. Given that MUC1 was specifically and significantly overexpressed in MT/Shc$^{313F/313F}$ breast cancer cells (Supplementary Data 1), we examined the relationship between MUC1 and STAT3 expression. Elevated MUC1 levels in Shc313F-expressing cells are ablated by STAT3 deletion (Fig. 4d).

We next injected these cells into the mammary fat pads of CD8$^{+/+}$ or CD8$^{-/-}$ mice, to examine the functional significance of STAT1 and STAT3 in modulating the balance between pro- versus anti-tumour immunity. STAT3 loss in MT/ShcA$^{+/+}$ breast cancer cells severely impaired tumour incidence in CD8$^{+/+}$ mice (100% versus 25% tumour-bearing mice) (Fig. 4e). In contrast, STAT3 deficiency had no impact on tumour development in CD8$^{-/-}$ animals (Fig. 4f). Histological

assessment of the mammary glands injected with MT/ShcA$^{+/+}$ breast cancer cells and their corresponding STAT1- or STAT3-deficient counterparts revealed the presence of microscopic lesions in $\sim$40% of the mammary glands at necropsy (Supplementary Fig. 8a). However, STAT3 is dispensable for subsequent tumour growth, irrespective of CD8 status (Fig. 4e,f). In contrast, STAT3 loss in MT/Shc$^{313F/313F}$ mammary tumours did not affect tumour onset but delayed the growth of established tumours, specifically in a CD8$^{+/+}$ background (Fig. 4e,f). Thus, STAT3 contributes, in part, to the establishment of an immunosuppressive microenvironment in MT/Shc$^{313F/313F}$ tumours. Emerging STAT3-null tumours re-established immune suppression based on the absence of a robust Granzyme B$^{+}$ response (Supplementary Fig. 8b,c) and impaired nuclear STAT1 translocation (Supplementary Fig. 9a). Loss of STAT3 in the epithelial component was confirmed by immunohistochemistry (Supplementary Fig. 9b,c).

We also examined the importance of STAT1 in tumours that retain high (MT/ShcA$^{+/+}$; MT/Shc$^{313F/313F}$) versus low (MT/Shc$^{2F/2F}$) STAT3 signalling. STAT1 deficiency in MT/Shc$^{313F/313F}$ mammary tumours had no impact on tumour onset or growth (Fig. 4e,f). Thus, chronic STAT1 activation in breast cancer cells selects for the activation of compensatory immunosuppressive signals. Surprisingly, STAT1 deletion in MT/ShcA$^{+/+}$ (STAT3$^{High}$) mammary tumours significantly impaired tumour onset in CD8$^{+/+}$ mice ($\sim$40% penetrance) but was dispensable for tumour initiation in CD8$^{-/-}$ animals, suggesting that STAT1 contributes to the establishment of immune suppression in STAT3$^{High}$ tumours (Fig. 4e,f). In contrast, STAT1 deficiency in STAT3$^{Low}$ mammary tumours (MT/Shc$^{2F/2F}$) significantly accelerated their growth (Fig. 4e), suggesting that STAT1 selectively confers an immune surveillance response in mammary tumours with low STAT3 activity. Combined, these data suggest that the STAT3 activation status of tumours may dictate whether STAT1 promotes or attenuates anti-tumour immunity, and that the ShcA pathway is an important integrator of this process.

To understand how STAT1 may serve an immunosuppressive role, we measured PD-L1 expression levels, which dampens anti-tumour immunity by activating the PD1 immune checkpoint on activated T cells[36]. STAT1 deficiency ablates IFN$\gamma$-stimulated PD-L1 levels in MT/ShcA$^{+/+}$ and MT/Shc$^{313F/313F}$ breast cancer cells (Fig. 5a). Moreover, increased PD-L1 expression levels are also observed in MT/Shc$^{2F/2F}$ orthotopic and transgenic mammary tumours (Fig. 5b,c). Finally, the ability of MT/Shc$^{2F/2F}$ tumours to induce PD-L1 expression is profoundly dependent on the presence of an intact immune response (Fig. 5c). These data suggest that PD-L1 is a STAT1 transcriptional target induced by the tumour microenvironment to facilitate eventual immune evasion. Our studies provide the first evidence that perturbing ShcA signalling has the potential to modulate immune responsiveness by altering the balance between STAT1-driven immune surveillance and STAT1/STAT3-driven immune suppression.

**ShcA pathway controls sensitivity to diverse immunotherapies.** To evaluate the clinical impact of these findings, we asked whether the pTyr-ShcA status of mammary tumours could impact their sensitivity to PD1 immune checkpoint inhibitors. We show that MT/ShcA$^{+/+}$ tumours are modestly sensitive to PD1 blockade. In contrast, MT/Shc$^{2F/2F}$ mammary tumours, which are impaired in STAT3 immunosuppressive signals, are exquisitely sensitive to PD1 immune checkpoint inhibition (Fig. 6a). Finally, selective loss of pY313-ShcA signalling exposes breast cancer cells to basally enhanced STAT1 and STAT3 signalling, to favour immune suppression. These tumours are

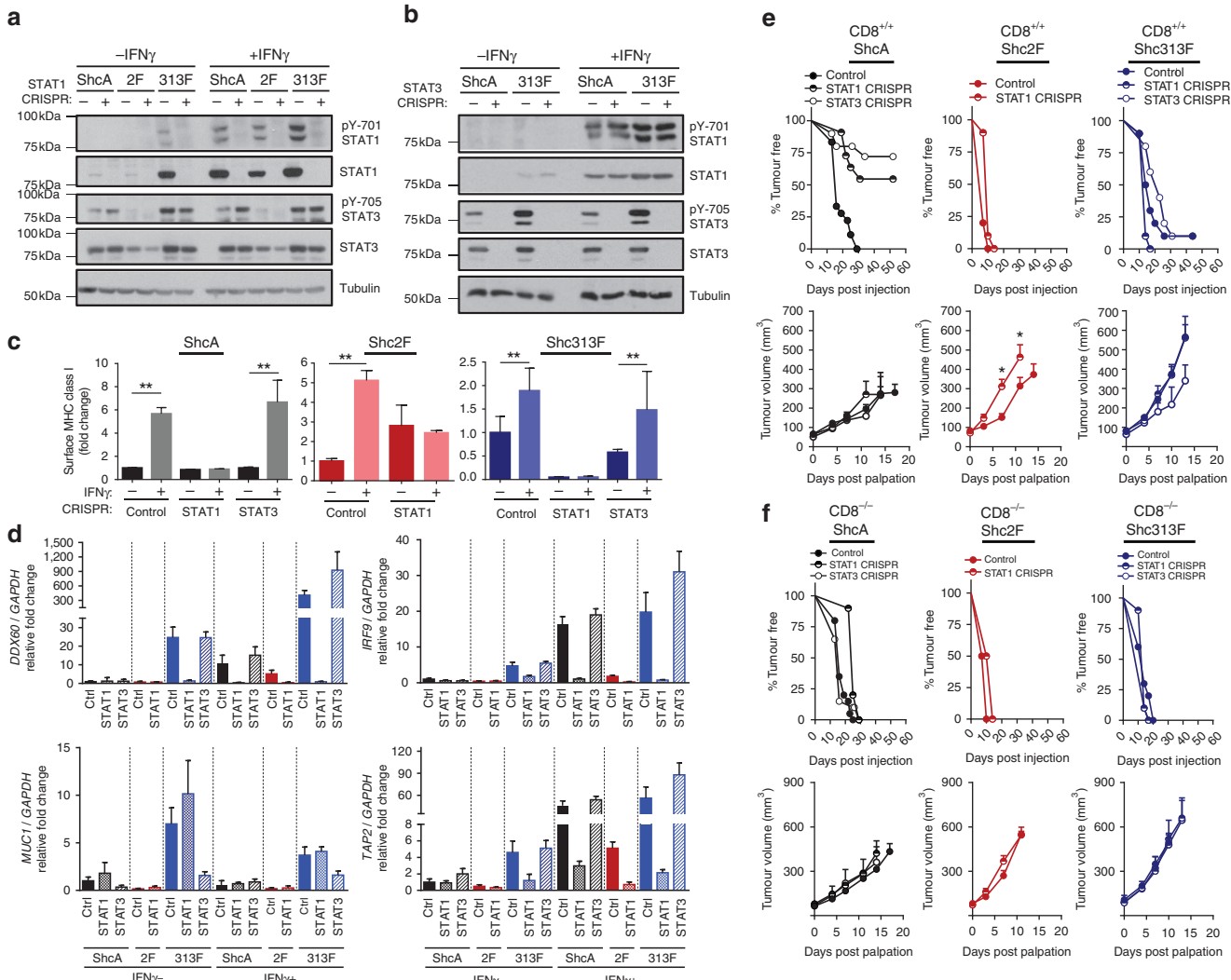

**Figure 4 | The STAT3 activation status of mammary tumours dictates whether STAT1 elicits pro- or anti-tumour immune responses.** MT/ShcA$^{+/+}$ (864), MT/Shc$^{2F/2F}$ (5372) and MT/Shc$^{313F/313F}$ (6738) cell lines were stably deleted of (**a**) *STAT1*, and (**b**) MT/ShcA$^{+/+}$ (864) and MT/Shc$^{313F/313F}$ (6738) cells were stably deleted of *STAT3* by CRISPR/Cas9 gene editing. Total cell lysates from the indicated cell lines were analysed by immunoblotting after 24 h of PBS or IFNγ (0.2 ng ml$^{-1}$) treatment. Representative image of three independent experiments is shown. (**c**) Surface MHC class I expression levels in the indicated cell lines as assessed by flow cytometry after 24 h IFNγ treatment (0.2 ng ml$^{-1}$) or PBS (control). Data represented as fold change in the geometric mean ± s.d. relative to control cell lines for each genotype ($n = 6$, two independent experiments). (**d**) Relative *IRF9*, *DDX60*, *TAP2* and *MUC1* mRNA levels, normalized to *GAPDH* levels in the absence (PBS) or presence (0.2 ng ml$^{-1}$) of IFNγ. The data are shown as the average fold change relative to PBS-treated MT/ShcA$^{+/+}$ (Control CRISPR) cells ± s.d. ($n = 5$ per group). (**e,f**) Mammary fat pad injection of the indicated cell lines into (**e**) FVB (CD8$^{+/+}$) or (**f**) CD8$^{-/-}$ mice. Tumour incidence is based on the per cent tumour-free mammary glands over the indicated days post injection. The rate of tumour outgrowth is represented as mean tumour volume (mm$^3$) ± s.e.m. ($n = 10$ tumours) (*$P < 0.05$ and **$P < 0.01$; unpaired two-tailed Student's *t*-test for **c**,**d**; one-way analysis of variance with Holm–Sidak method for **e**,**f**).

insensitive to the anti-tumorigenic effects of PD1 immune checkpoint blockade (Fig. 6a). Considering the chronically elevated STAT1 activation status of MT/Shc$^{313F/313F}$ tumours, we reasoned that they could be sensitized to alternative immunotherapeutic strategies that favour re-activation of antigen-specific tumour immunity. To test this, FVB mice were vaccinated with mitotically arrested MT/ShcA$^{+/+}$, MT/Shc$^{2F/2F}$ or MT/Shc$^{313F/313F}$ breast cancer cells before mammary fat pad injection. Tumour vaccination strategies modestly impacted the tumorigenic potential of MT/ShcA$^{+/+}$ tumours (30% tumour free) and ablated MT/Shc$^{2F/2F}$ tumour initiation (100% tumour free) even 3 months post tumour cell inoculation (Fig. 6b). Although MT/Shc$^{313F/313F}$ mammary tumours are refractory to the effects of PD1 immune checkpoint blockade, they are

exquisitely sensitive to tumour vaccination strategies, whereby 80% of animals remained tumour free 90 days post inoculation (Fig. 6b). Comparison of surface MHC class I expression levels on each cell line used for immunization shows that MHC class I levels are comparable between MT/ShcA$^{+/+}$ and MT/Shc$^{2F/2F}$ breast cancer cells, even though only the latter is exquisitely sensitive to tumour vaccination. In contrast, MT/Shc$^{313F/313F}$ breast cancer cells basally upregulate surface MHC class I expression levels relative to MT/ShcA$^{+/+}$ cells (Fig. 2f,g), which may contribute, in part to their increased sensitivity to immunization strategies.

Combined, these studies suggest that an intact ShcA pathway renders tumours insensitive to the effects of several immunotherapies. We suggest that selective loss of pY239/240-ShcA

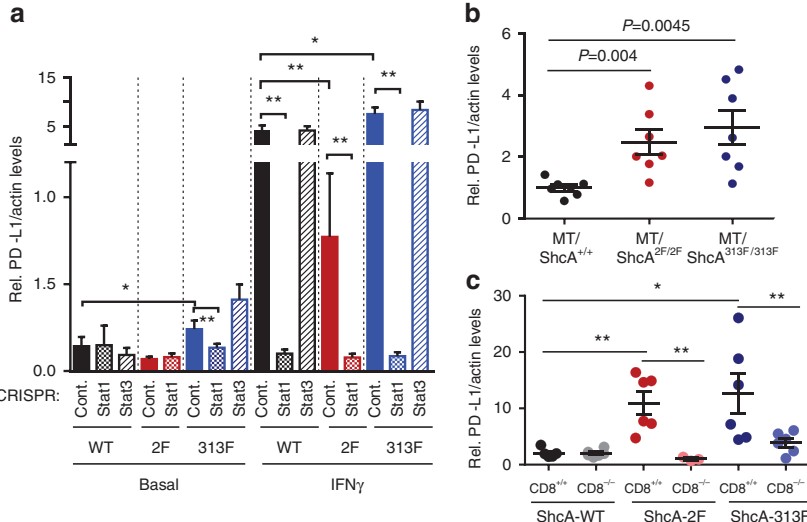

**Figure 5 | Loss of phospho-Y239/240-ShcA signalling increases PD-L1 expression on mammary tumours.** (**a**) Relative *PD-L1* mRNA levels normalized to *Actb* levels were determined by RT–quantitative PCR in control, *STAT1*-null or *STAT3*-null cells of the indicated genotypes: MT/ShcA$^{+/+}$ (864), MT/Shc$^{2F/2F}$ (5372), MT/Shc$^{313F/313F}$ (6738)) that were cultured in the absence (PBS) or presence (0.2 ng ml$^{-1}$) of IFNγ for 24 h. Data represented as mean ± s.d. (**b**) Relative *PD-L1* mRNA levels (normalized to *Actb*) in mammary tumours derived from MT/ShcA$^{+/+}$, MT/Shc$^{2F/2F}$ and MT/Shc$^{313F/313F}$ transgenic mice (n = 7 tumours per genotype) ± s.e.m. (**c**) Relative *PD-L1* mRNA levels (normalized to *Actb*) in mammary tumours from CD8$^{+/+}$ and CD8$^{-/-}$ mice injected with breast cancer cells of the indicated genotypes: MT/ShcA$^{+/+}$ (864), MT/Shc$^{2F/2F}$ (5372) and MT/Shc$^{313F/313F}$ (6738). Data represented as mean ± s.e.m. (n = 6 tumours each) (\*P < 0.05 and \*\*P < 0.01; unpaired two-tailed Student's *t*-test for **a** and Wilcoxon's rank-sum test for **b,c**).

signalling dampens STAT3 immunosuppressive signals and renders tumour cells sensitive to immune checkpoint inhibitors and tumour vaccines. In contrast, specific loss of pY313-ShcA signalling potentiates STAT1 signalling, in the context of elevated STAT3 activation. The increased immunosuppressive environment of MT/Shc$^{313F/313F}$ tumours renders them insensitive to immune checkpoint blockade but their elevated STAT1 status may be therapeutically exploited to sensitize them to vaccination-based therapies (Fig. 6c).

**ShcA activity and immune evasion in human breast cancer.** To elucidate the significance of these observations to human breast cancer, we generated gene signatures using the differentially expressed genes obtained from our RNAseq analyses. After stratification of genes with human orthologues, we generated the following signatures: common differentially expressed (aka: double mutant), Shc2F specific and 313F specific (Supplementary Data 1). We performed single-sample gene set enrichment analysis (ssGSEA), a computational method that calculates the absolute degree of enrichment of a gene set in individual samples[37], to assess the presence of each signature in human breast cancers from The Cancer Genome Atlas (TCGA) data set (n = 1,215). We examined the correlation of the three gene signatures with expression levels of *CD8A*, *GZMB* and *PD-L1* (Fig. 6d). We observe that *GZMB* mRNA levels positively correlate with Shc2F-like and Shc313F-like gene signatures (Spearman's correlation R = 0.42 and R = 0.39, respectively). A Shc313F-like signature displays a modestly higher correlation with *CD8A* (R = 0.26) and *PD-L1* levels (R = 0.28) compared with tumours with an enhanced 2F-like response (R = 0.22 and R = 0.15, respectively). A common double mutant ShcA gene signature did not associate with changes in *CD8A*, *GZMB* and *PD-L1* levels (R = 0.005, R = 0.007 and R = 0.13) (Fig. 6d), suggesting that unique signalling pathways downstream of the Y239/240- or Y313-ShcA phospho sites contribute to immune suppression.

We next interrogated whether our gene signatures could stratify STAT1 or STAT3 transcriptional responses in human breast

tumours. We generated STAT1 and STAT3 gene signatures (Supplementary Tables 2 and 3), which were also subjected to ssGSEA analysis. We first verified that the STAT1 ssGSEA signature positively correlated with relative STAT1 protein levels in independent MT/ShcA$^{+/+}$, MT/Shc$^{2F/2F}$ and MT/Shc$^{313F/313F}$ cell lines (Supplementary Fig. 10a). We also demonstrate that the STAT3 ssGSEA signature positively correlates with increased Y705-STAT3 phosphorylation in each cell line (Supplementary Fig. 10b). These gene signatures correlate, as expected, with the mRNA levels of their respective transcription factors (STAT1, Spearman's R = 0.89; STAT3, Spearman's R = 0.45) and with the phospho-protein levels (pY705-STAT3: Spearman's R = 0.37; pY701-STAT1: data not available) in human breast tumours from the TCGA data set (Supplementary Fig. 11). We show that a Shc313F-like gene signature strongly correlates with a STAT1 ssGSEA score (R = 0.61), whereas STAT3 ssGSEA scores are anti-correlated with a Shc2F-like signature (R = −0.16) (Fig. 6d). Finally, we stratified tumours from the TCGA data set based on their STAT1 and STAT3 activation status using the aforementioned ssGSEA scores (Fig. 6e). Comparable numbers of primary breast tumours segregated into one of four categories: STAT1$^{Low}$ (first quartile)/STAT3$^{Low}$ (first quartile), n = 110 or (9.1%); STAT1$^{Low}$ (first quartile)/STAT3$^{High}$ (fourth quartile), n = 58 (4.8%); STAT1$^{High}$ (fourth quartile)/STAT3$^{Low}$ (first quartile), n = 61 (5%); or STAT1$^{High}$ (fourth quartile)/STAT3$^{High}$ (fourth quartile), n = 91 (7.5%). We show that an elevated STAT1 transcriptional response is associated with a robust increase in *GZMB* (13-fold), *CD8A* (4.3-fold) and *PD-L1* (2.6-fold) expression levels in STAT3$^{Low}$ tumours but not in STAT3$^{High}$ tumours (1.9-fold, 1.4-fold and 1.1-fold, respectively) (Fig. 6e). These data support our observation that the ability of breast tumours to mount effective STAT1 anti-tumour immune responses is dependent on reduced STAT3 activation. They further suggest that the STAT1/STAT3 ratio is dynamically regulated in individual human breast tumours and support the potential for utilizing this ratio to predict intrinsic immune surveillance and sensitivity of breast tumours to specific immunotherapies.

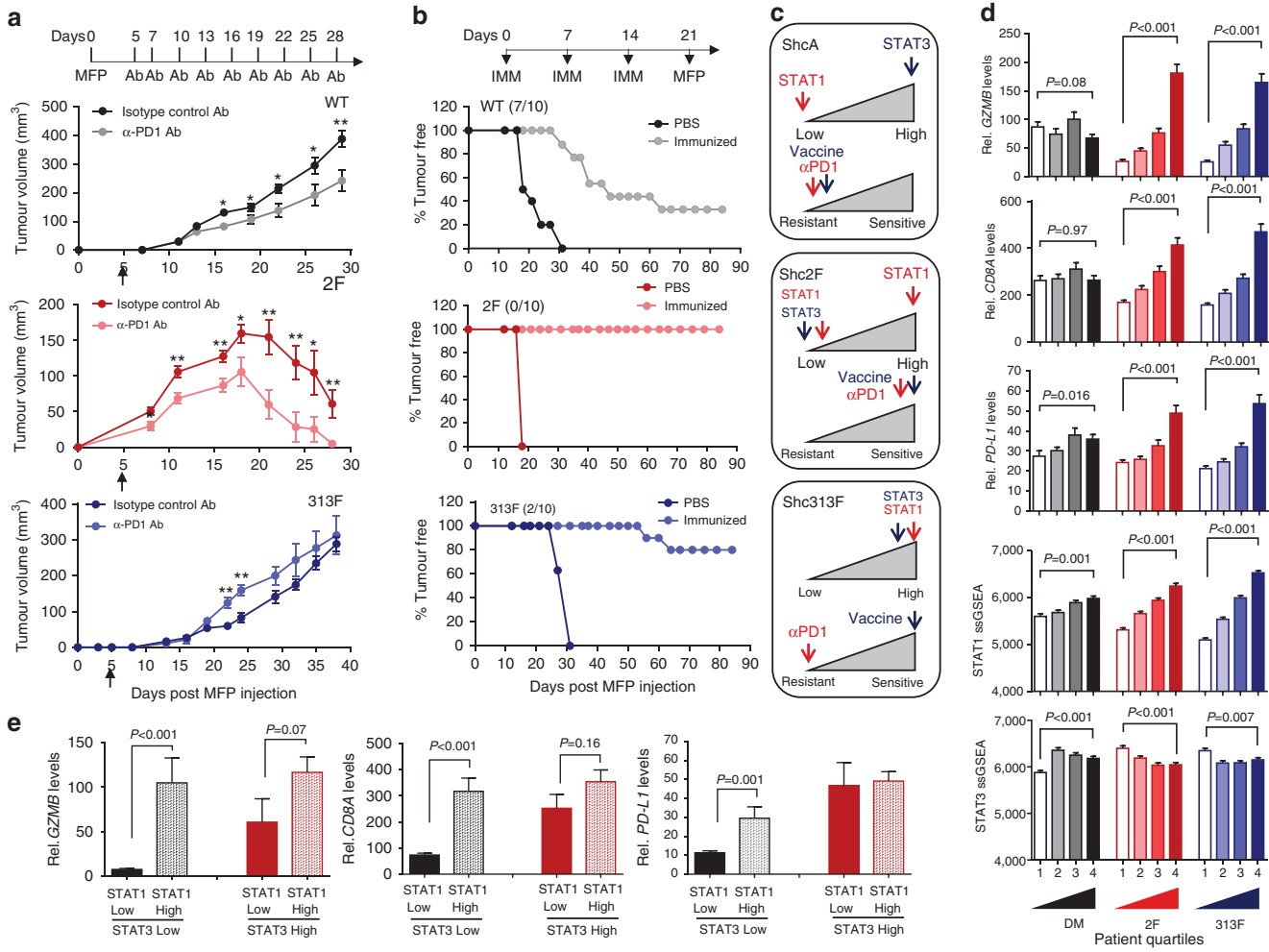

**Figure 6 | Distinct ShcA signalling networks differentially sensitize mammary tumours to immunotherapies.** (**a**) Mammary fat pad injection of FVB mice with the indicated cell lines. Starting on day 5, mice were treated with 100 μg of a neutralizing PD1 (α-PD1) antibody or its corresponding isotype control IgG and every 3 days thereafter (n = 10 tumours each). Data represented as mean ± s.e.m. (**b**) FVB mice received three intraperitoneal injections (days 0, 7 and 14) with PBS or mitomycin C-treated breast cancer cells of the indicated genotypes. On day 21, mammary fat pad injections were performed with breast cancer cells of the same genotype used for vaccination (n = 9–11 mice each). (**c**) Schematic diagram illustrating the relationship between ShcA-driven, STAT1/STAT3 activation and sensitivity to PD1 immune checkpoint inhibitors or tumour vaccination strategies. (**d**) Primary breast tumours from the TCGA RNAseq data set (n = 1,215) were equally stratified into four quartiles based on gene expression signatures that are either unique to loss of the Y239/240 (2F) or Y313 phosphorylation site (313F), or shared in both groups (double mutant, DM). ssGSEA was used to rank order each tumour based on acquisition of a DM, 2F or 313F-like ShcA signature. Tumours in the first quartile resemble those that possess elevated phosphotyrosine-dependent ShcA signalling, whereas those in the fourth quartile are reminiscent of the lowest degree of ShcA-dependent transcriptional responses. The average GZMB, CD8A and PD-L1 mRNA levels were evaluated in each quartile. The same tumours were stratified based on relative expression levels of STAT1 or STAT3 target genes. The average STAT1 and STAT3 ssGSEA levels were determined for tumours in each quartile. The data are shown as average expression levels ± s.e.m. (comparing quartiles 1 and 4). (**e**) Tumours (n = 320) were stratified by STAT1[Low] (first quartile)/STAT3[Low] (first quartile), n = 110 or (34.4%); STAT1[Low] (first quartile)/STAT3[High] (fourth quartile), n = 58 (18.1%); STAT1[High] (fourth quartile)/STAT3[Low] (first quartile), n = 61 (19%); or STAT1[High] (fourth quartile)/STAT3[High] (fourth quartile), n = 91 (28.4%) ssGSEA signatures. Relative GZMB, CD8A and PD-L1 expression levels are plotted ( ± s.e.m.). Significance was determined using multiple t-test with Holm–Sidak method for **a** (*P < 0.05 and **P < 0.01) and unpaired two-tailed Student's t-test for **e**,**d**.

## Discussion

This study provides novel mechanistic insight into how tyrosine kinases engage ShcA to promote immune suppression. We show that tyrosine kinases engage pY239/240-ShcA to sustain STAT3 immunosuppressive signals and simultaneously employ pY313ShcA to restrain activation of STAT1-driven anti-tumour immunity. This does not preclude the possibility that other ShcA-coupled, STAT-independent signalling pathways contribute to the observed immune phenotypes. Our genetic approaches functionally demonstrate that the CD8[+] T-cell compartment is clearly essential for the induction of anti-tumour immune responses in MT/Shc[2F/2F] tumours. However, we cannot exclude

the possibility that natural killer cells also contribute to heightened immune surveillance.

Although the ShcA mutations employed in this study (Y239/240F and Y313F) are not found in human breast cancer, they provide a valuable genetic tool to interrogate the mechanistic basis for how perturbations in ShcA signalling affect immune suppression. Significant experimental evidence show that ShcA expression and activity varies widely among individual breast cancers and is relevant to patient outcome. For example, ShcA represents a key convergent point downstream of tyrosine kinases that are important for breast cancer development. Moreover, ShcA protein levels vary widely across individual breast cancers,

are enriched in the HER2 and basal subtypes, and associate with inferior clinical outcome[6]. This is consistent with the fact that ShcA resides within an amplicon (Chr1q21–23) that is observed in 15% of all breast cancers, particularly in basal and luminal/p53 negative tumours[38]. Previous studies have also shown that increased ShcA tyrosine phosphorylation in primary breast cancers correlates with lymph node status, tumour stage and recurrence[39]. In combination, these studies provide a solid rationale for understanding the molecular basis by which ShcA transduces oncogenic signals that promote breast cancer immune suppression.

Our genetic approach to manipulating ShcA-coupled signalling pathways downstream of individual ShcA phosphorylation sites (Y239/240F or Y313F) phenocopies breast tumours that are intrinsically reduced in tyrosine kinase/ShcA signalling or inducibly repress the ShcA signalling axis in response to tyrosine kinase inhibitors. The genotypes employed in this study allow us to model three states: intrinsic immune suppression (MT/ShcA$^{+/+}$), intrinsic immune surveillance (MT/ShcA$^{2F/2F}$) and acquired immune suppression (MT/ShcA$^{313F/313F}$). MT/ShcA$^{+/+}$ cells model breast tumours that possess an activated tyrosine kinase/ShcA axis, which simultaneously activates STAT3 and represses STAT1 to establish immune suppression. In contrast, MT/Shc$^{2F/2F}$ tumours represent those human breast cancers that are low in tyrosine kinase/ShcA signalling (intrinsic or therapy induced). This genetic tool provides new knowledge that suppression of STAT3 signalling downstream of pY239/240-ShcA selectively increases immune surveillance. Finally, MT/ShcA$^{313F/313F}$ cells represent tumours that were debilitated in tyrosine kinase/ShcA signalling but that hyperactivated compensatory signalling pathways (both STAT3 dependent and independent) to re-acquire immune suppression (Fig. 7).

In combination, these studies suggest that the development of inhibitors against the Y239/240-ShcA phosphorylation sites may represent a therapeutic strategy to inhibit STAT3 activation in cancer cells and increase sensitivity to immunotherapies. Given the pleotropic mechanism of action of ShcA in potentiating breast cancer progression, ShcA-targeted therapies would further elicit additional tumoricidal responses including, but not limited to, impaired angiogenesis in breast tumours to increase the likelihood of therapeutic success[18,19].

We propose that the strong immunosuppressive state imposed by Shc313F-like breast tumours (STAT1$^{High}$/STAT3$^{High}$) limits the therapeutic potential of strategies that aim to block T-cell peripheral tolerance, such as PD1 immune checkpoint blockade. Indeed, we could identify human breast tumours that coordinately increase STAT1- and STAT3-driven transcriptional responses in human breast cancers (Fig. 6). We provide the first experimental evidence that an elevated STAT1 response in these tumours may sensitize them to immunotherapeutic approaches that serve to augment STAT1-driven anti-tumour immune responses, including vaccine-based strategies. These studies demonstrate that the STAT1/STAT3 ratio may represent a useful biomarker to not only predict the degree of breast cancer immune suppression but also the type of immunotherapy that is most likely to yield durable clinical responses in individual breast tumours (Fig. 7).

A recent study provided the first experimental evidence that STAT3 is essential for the establishment of immunosuppression during the earliest stages of breast cancer progression[34]. This is consistent with our observations that mammary epithelial STAT3 loss profoundly impairs breast cancer development in immunocompetent, but not immunodeficient, mice. Considering these observations, STAT3 represents an attractive target for therapeutic intervention. STAT3 decoy oligonucleotides

decreased tumour growth in pre-clinical models[40], which led to phase I clinical trials in solid tumours, including breast cancer. STAT3 inhibition in numerous cell types within the immune microenvironment has the potential to significantly alleviate tumour immune suppression. For example, inhibition of STAT3 signalling in immune cell types with anti-tumorigenic properties, including natural killer cells and cytotoxic T lymphocytes, significantly increased their tumoricidal properties in numerous cancer types[41–43]. Moreover, increased STAT3 signalling in dendritic cells inhibited their maturation and subsequent ability to serve as antigen-presenting cells in educating anti-tumour immune responses[44]. However, other studies suggest that STAT3 signalling in immune cells is also important for the initiation of inflammatory responses, which are essential to initiate and educate anti-tumour T-cell responses[45]. Moreover, in regulatory T cells, STAT3 is engaged downstream of inflammatory cytokine signalling, such as the interleukin-6 receptor, to limit their immunosuppressive properties[46]. Our study suggests that inhibition of STAT3 signalling, specifically in the tumour epithelium, robustly engages immune surveillance. We suggest that the development of inhibitors that bind the Y239/240-ShcA phosphorylation sites may represent an alternative therapeutic strategy to inhibit STAT3 activation in breast cancer cells.

Conflicting literature supports both pro- and anti-tumorigenic roles for STAT1 in breast cancer development. Several studies suggest that tumour-intrinsic STAT1 functions as a tumour suppressor through its ability to upregulate interferon sensitive genes that promote cell cycle arrest, APP or activation of inflammatory responses[47,48]. Indeed, pre-clinical studies in ErbB2-driven transgenic mouse models demonstrate that mammary epithelial STAT1 loss accelerates tumour development[49,50]. This observation is supported by an independent study, which demonstrates that STAT1-null animals spontaneously develop ERα+ breast cancer[51].

However, numerous studies also suggest that STAT1 induces tumour intrinsic and stromal gene expression changes that promote breast cancer development. Increased STAT1 activity in breast tumours promotes immune suppression by mobilizing myeloid-derived suppressor cells in human breast cancers[52]. STAT1 also directly controls immune suppression by upregulating the expression of PD-L1 on tumour cells. For example, increased STAT1 activity in therapy-resistant endometrial cancers correlates with increased PD-L1 expression levels[53]. Our data suggests that elevated STAT3 immunosuppressive signals enable persistent STAT1 activation by limiting its anti-tumour responses and thereby potentiating STAT1-driven, PD-L1-dependent induction of T-cell tolerance. This is further supported by our observation in primary human breast cancers that elevated STAT1 transcriptional responses are only associated with enhanced *PD-L1* expression levels in tumours that possess a low STAT3 transcriptional signature. Several studies further suggest that STAT1 plays an important role in the development of therapeutic resistance. Increased STAT1 signalling has been observed in endocrine-resistant breast cancers[54]. STAT1 also increases cap-independent translation of proteins that promote cell survival to increase chemoresistance[55].

Recent literature supports the combination of tyrosine kinase inhibitors with immunotherapies. For example, PD-L1 blockade increases the efficacy of B-cell lymphomas to Ibrutinib, a dual BTK/ITK tyrosine kinase inhibitor[56]. Moreover, Lapatinib, an EGFR/ErbB2 tyrosine kinase inhibitor, potentiates STAT1-dependent activation of immune surveillance[57]. However, our data suggests that tyrosine kinase inhibitors may establish a cellular state that is hard-wired to facilitate eventual immune evasion. Indeed, tyrosine kinase inhibitors will impair both Y239/240 and Y313-ShcA phosphorylation to limit STAT3

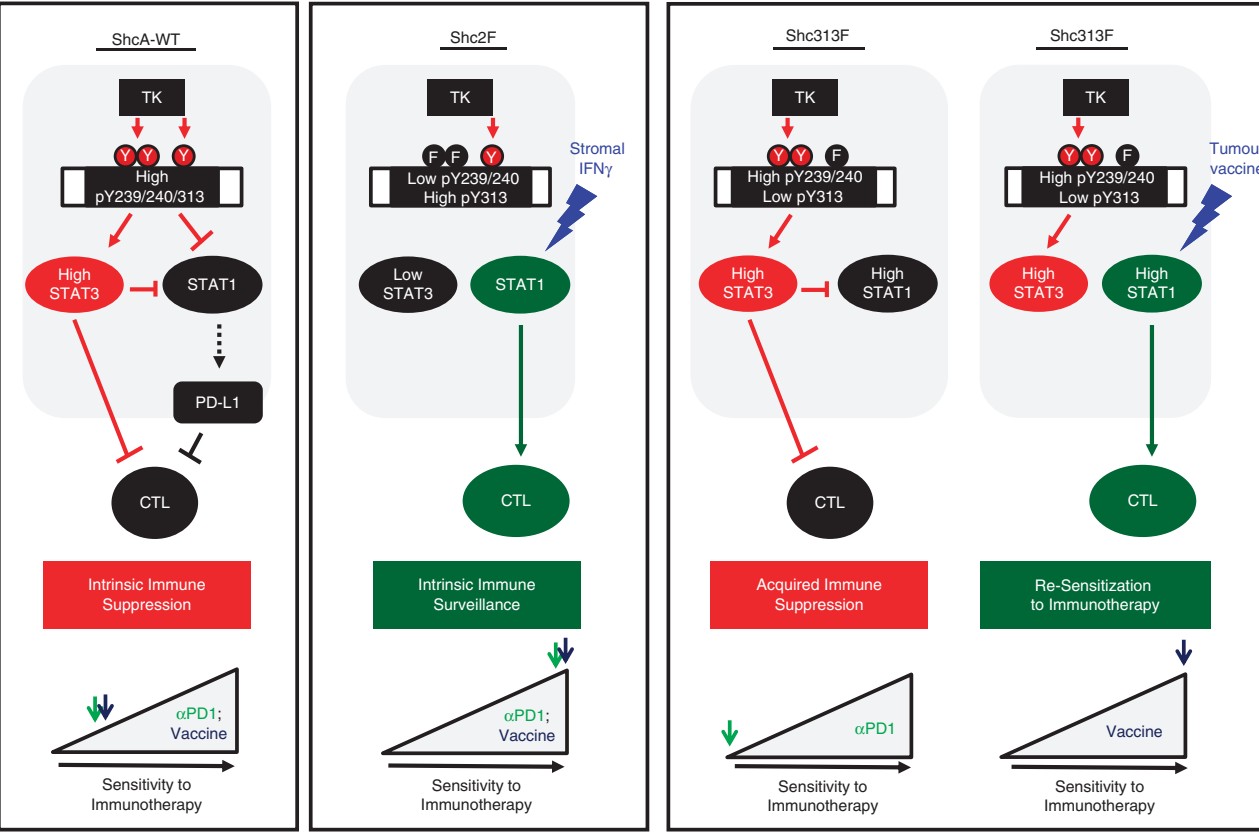

**Figure 7 | Schematic representation outlining the role of how modulating the TK/ShcA axis impacts breast cancer immune suppression.** Elevated STAT3 signalling promotes immune suppression in ShcA-WT breast tumours, which suppresses STAT1-driven anti-tumorigenic immune responses. Instead, STAT1 further contributes to immune evasion by increasing PD-L1 levels in mammary tumours. As such, these tumours display a modest sensitivity to PD1 immune checkpoint blockade or tumour vaccination strategies. In contrast, loss of the phospho-pY239/240 ShcA signalling (Shc2F) significantly and specifically impairs STAT3 activation in breast cancer cells. This relieves STAT3-driven immune suppression, leading to stromally induced activation of STAT1-mediated anti-tumorigenic responses in mammary tumours, which together promote immune surveillance and significant responsiveness to anti-PD1 therapies or tumour vaccines. Finally, specific loss of pY-313 ShcA signalling (313F) directly increases STAT1 signalling in breast cancer cells, leading to a compensatory hyperactivation of STAT3 signalling. Heightened STAT3 signalling in 313F mammary tumours sustains immune evasion, leading to increased resistance to PD1 checkpoint inhibitors. Paradoxically, however, increased STAT1 signalling in these tumours increases their sensitivity to tumour vaccination strategies, owing to the heightened increase in baseline STAT1 signalling.

immunosuppressive signals and activate STAT1 in breast cancer cells. This elevated STAT1 response may render tumours susceptible to the re-establishment of immune evasion following selective STAT3 re-activation.

In addition to tyrosine kinase inhibitors, several studies suggest that re-activation of local immune responses is an essential component to sensitize solid tumours to immunotherapies. For example, induction of inflammatory responses by chemotherapies increases the responsiveness of lung adenocarcinomas to immune checkpoint inhibitors[58]. Beyond these inducible responses, a robust body of literature supports the fact that a high mutational load in cancer cells facilitates the emergence of neo-antigens to activate T cells that are not susceptible to immune tolerance. This is particularly relevant to metastatic melanoma, which harbours high mutational loads and shows the greatest clinical benefit to immunotherapies[59,60].

Numerous studies, however, also support a role for non-genomic alterations in cancer cells that dictate their sensitivity to immunotherapies. For example, epigenetic silencing of IFN-stimulated cytokines limits the sensitivity of ovarian tumours to PD-L1 immune checkpoint blockade or adoptive T-cell-based immunotherapies[61]. Moreover, treatment of colorectal cancers with DNA demethylating agents induces the

expression of numerous IFN-responsive genes to mimic a virally infected state and induce an anti-tumour immune response[62]. We provide the first experimental evidence that posttranslational modification of specific signal transduction pathways downstream of tyrosine kinases also has an impact on sensitivity to immunotherapies in breast cancer. We suggest that the development of inhibitors that prevent Y239/240-ShcA phosphorylation or that competitively bind the Y239/240-ShcA phospho site may sensitize breast tumours to immunotherapies. These observations are timely given the interest in both tyrosine kinase inhibitors and immunotherapies in breast cancer clinical trials.

## Methods

**Mice.** MMTV/MT[63] and MMTV/NIC transgenic mice[19] have been described. Mice expressing a *ShcA^{flx}* allele or mutant ShcA alleles harbouring tyrosine-to-phenylalanine point mutations at residues 239/240 (Shc2F) and 313 (Shc313F), under the control of the endogenous promoter have been described[29]. All mouse strains are on a pure FVB background. CD8$^{-/-}$ and IFN$\gamma^{-/-}$ mice were purchased from Charles River Laboratories and backcrossed onto an FVB background for eight generations before initiating this study. Transgenic mice were monitored for tumour onset by weekly physical palpation. FVB mice were purchased from Charles River Laboratories.

For mammary fat pad injection, $5 \times 10^5$ breast cancer cells were injected into the fourth mammary fat pad of female mice (6–10 weeks of age). Animals were

monitored for tumour growth biweekly via caliper measurements. For the *in vivo* immunization experiment, breast cancer cells were mitotically arrested with 225 μM mitomycin C (Abcam) for 3 h and injected intraperitoneally into FVB mice ($1 \times 10^6$ cells) to generate an immunized cohort. Control groups were mock injected with PBS. This injection schedule was repeated 7 and 14 days later. On day 21, immunized and non-immunized mice were subjected to mammary fat pad injection with the cell line that was used for immunization. For the PD1 immune checkpoint blockade studies, animals were treated with 100 μg of a neutralizing α-PD1 antibody (clone RMP1-14, BioXCell) or its corresponding isotype control IgG (InVivoMAb Rat IgG2a, clone 2A3, BioXCell) 5 days post mammary fat pad injection and every 3 days thereafter. Tumour outgrowth was measured every 3 days by caliper measurements ($n = 10$ each). All animal studies were approved by the Animal Resources Council at McGill University and comply with guidelines set by the Canadian Council of Animal Care.

**Generation and culture of primary cell lines.** Tumours were surgically excised from transgenic mice, washed in ice-cold PBS and minced using a McIlwain tissue chopper (Campden Instruments). Tissues were incubated with pre-warmed DMEM medium (with penicillin and streptomycin) containing 2.4 mg ml$^{-1}$ Dispase (Roche; Neutral protease, grade II) and 2.4 mg ml$^{-1}$ Collagenase B (Roche) and incubated at 37 °C with gentle agitation for 2–3 h. Cells were washed three times in 1 mM EDTA/PBS followed by centrifugation at 800 r.p.m. for 3 min. Cells were resuspended in 2.5% fetal bovine serum (FBS)/DMEM growth media containing mammary epithelial growth supplement (MEGS; 3 ng ml$^{-1}$ human epidermal growth factor, 0.5 μg ml$^{-1}$ hydrocortisone, 5 μg ml$^{-1}$ Insulin and 0.4%v/v bovine pituitary extract). For experimental purposes, cells were cultured in 1% FBS/MEGS-containing DMEM and treated either with IFNγ (485-MI-100; R&D Systems) or PBS. Cell lines are routinely screened for mycoplasma contamination (either on a monthly basis or before any *in vivo* experiment).

**CRISPR/Cas9 genome editing.** Sequences targeting murine *STAT1* and *STAT3* were designed using the CRISPR Design Tool (http://crispr.mit.edu): *STAT1* 5′-GTACGATGACAGTTTCCCCATGG-3′ and 5′-GGACTCCAAGTTCCTGG AGCAGG-3′ within Exon 3. Targeted sequence for *STAT3* was 5′-GGAACT GCCGCAGCTCCATGGGG-3′ within Exon 1. The gBlocks containing U6 promoter, the designed target sequence, gRNA scaffold and termination signal were purchased from IDT. Clones verified to have lost expression of STAT1 or STAT3 by immunoblot analysis were pooled for subsequent analysis ($n = 6$). For the STAT1 CRISPR cohort, the pooled cells were derived from two individual guide sequences.

**Immunoblot, immunochemistry and ELISA.** Adherent cells and tumour tissues from transgenic mouse and mammary fat pad injections were lysed in PLCγ cell lysis buffer[19]. All the primary antibodies and their conditions used in this study are listed in Supplementary Table 4. All uncropped immunoblots are shown in Supplementary Fig. 13. CXCL9 protein levels in supernatants from cells following 24 h of IFNγ (1 ng ml$^{-1}$) or control (PBS) stimulation were determined using Mouse CXCL9/MIG DuoSet (DY492) ELISA kit (R&D Systems).

For immunohistochemical analyses, paraffin-embedded sections (4 μm) were stained[6]. Primary antibodies used along with their staining conditions are listed in Supplementary Table 5. Slides were scanned using a ScanScope XT Digital Slide Scanner (Aperio) and data were analysed using Image Scope software.

**Flow cytometry.** Spleens were homogenized in PBS using polypropylene pestles (1212M63; Thomas Scientific), filtered through 70 μm mesh cell strainer and further diluted to 5 ml total volume with PBS. Tumour tissues (approximately 400–500 mm$^3$) were dissociated as described above. Dissociated tumour cells were resuspended in 6 ml 0.3 mg ml$^{-1}$ DNase I (Sigma), incubated for 20 min at 37 °C, rinsed in cold 1 mM EDTA/PBS, filtered through 70 μm mesh cell strainer and diluted to 10 ml total volume with cold 2% FBS/PBS. Dissociated tumour cells ($2 \times 10^7$) or splenocytes ($2 \times 10^6$) were stained with Live/Dead Fixable Aqua 405 nm (catalogue number L34957, ThermoFisher). Samples were blocked in Fc block CD16/CD32 (catalogue number 553142, BD Biosciences) and stained with primary fluorescently conjugated antibodies (listed in Supplementary Table 6) for 30 min at 4 °C. Samples were analysed by LSR Fortessa Cell Analyzer (BD Biosciences) and FlowJo software. Aggregates were gated out using FSC-A versus FSC-H and SSC-A versus SSC-H, and total cells were selected. B220 was used to exclude B cells from the analysis. B220$^-$ cells were further subdivided into CD8$^+$CD69$^{+/-}$ to determine the percentage of CD8$^+$ cytotoxic T cells and CD8$^+$CD69$^+$ activated cytotoxic T-cells. The percentage of MDSCs was determined by selecting the immune cells from the live cell population using CD45. MDSCs were defined as CD45$^+$CD11b$^+$Gr1$^+$. All percentages were reported as per cent of total events analysed. To measure surface MHC class I expression levels, $2 \times 10^6$ breast cancer cells were stained for 30 min with 0.5 μg of phycoerythrin (PE) fluorophore-conjugated anti-mouse MHC class I antibody (H-2Db) (catalogue number 12-5999-83, eBioscience).

**RNA isolation, RT-qPCR and RNA sequencing.** Total RNA from cell lines was extracted using TRIzol reagent and that of tumour tissues was extracted using RNeasy Midi Kits (QIAGEN). Subsequent complementary DNA synthesis and RT–quantitative PCR analyses were carried out using the primers listed in Supplementary Table 7.

For the RNAseq studies, total RNA was isolated from ShcWT, Shc2F and Shc313F established MMTV/MT breast cancer cell lines ($n = 4$ per genotype) using RNeasy Mini Kits (QIAGEN). Cells were cultured in 1% FBS/MEGS media for 24 h. RNAseq was performed at the McGill University and Genome Quebec Innovation Centre. RNA quality was assessed by Agilent 2100 Bioanalyzer (Agilent Technologies). Libraries for RNAseq were prepared according to strand-specific Illumina TruSeq protocols. Samples were multiplexed at four samples per lane and sequenced on an Illumina HiSeq 2000 instrument (100 bp, paired-end reads).

**Bioinformatics—RNA sequencing analysis.** Sequencing reads were trimmed using Trimmomatic v0.32 (ref. 64), removing low-quality bases at the ends of reads (phred33 < 30) and clipping the first four bases in addition to Illumina adaptor sequences using palindrome mode. A sliding window quality trimming was performed, cutting once the average quality of a window of four bases fell below 30. Reads shorter than 30 bp after trimming were discarded. The resulting high-quality RNAseq reads were aligned to the mouse reference genome build mm10 using STAR v2.3.0e[65]. Uniquely mapped reads were quantified using featureCounts v1.4.4 and the UCSC gene annotation set. Integrative Genomics Viewer was used for visualization. Multiple quality control metrics were obtained using FASTQC v0.11.2, SAMtools[66], BEDtools[67] and custom scripts.

*RNAseq gene expression analysis.* Global expression changes were assessed by unsupervised hierarchical clustering of samples and principal component analysis (PCA). To this end, expression levels were estimated using exonic reads mapping uniquely within the maximal genomic locus of each gene and its known isoforms. Normalization (median of ratios) and variance stabilized transformations of the data were performed using DESeq2 (ref. 68). Pearson's correlation was used as the distance metric for hierarchical clustering and average linkage as the agglomeration method. Bootstrapped hierarchical clustering was computed using the R package pvclust[69]. Differential expression analysis to identify expression changes with respect to wild-type (WT) ShcA controls was performed using DESeq2 (ref. 68). Genes with statistically significant (adjusted *P*-value < 0.05) and large (fold change > 2) expression changes, expressed above a threshold (average normalized expression across samples > 100) were selected to derive gene signatures associated with each genotype. Human leukocyte antigen genes, genes with no known function and genes with no human orthologues were removed from downstream analyses. To acquire the ShcA-regulated gene signatures, we first compared genes that are differentially expressed between the following groups: (1) ShcA-WT versus Shc2F and (2) ShcA-WT versus Shc313F. We then compared both lists of differentially expressed genes to identify: (a) genes that are commonly differentially expressed in all Shc2F cell lines relative to the rest (Shc2F-like), (b) genes that are commonly differentially expressed in all Shc313F cell lines relative to the rest (Shc313F-like) and (c) genes that are commonly differentially expressed in both Shc2F and Shc313F cells relative to ShcA-WT cells.

*STAT1* and *STAT3* gene signatures, on the other hand, were derived from previously reported validated targets (Supplementary Tables 2 and 3). In addition, we required that mRNA levels of these across patient samples displayed a Spearman's correlation R > 0.1 with *STAT1* and *STAT3* mRNA levels, respectively.

All gene signatures were projected across 1,215 human breast cancers from TCGA data set using ssGSEA as described before[37]. Briefly, a score is defined to represent the degree of enrichment of a given gene set in a sample: gene expression values for each sample are rank-normalized and an enrichment score is produced using the empirical cumulative distribution functions (ECDF) of genes, with the final score computed by integrating the difference between a weighted ECDF of genes in the signature and the ECDF of the remaining genes[37]. This calculation is repeated for each signature and each sample in the data set. To compute ssGSEA scores, we used the GenePattern software implementation from the Broad Institute, ssGSEAProjection (v6)[70]. We first verified that the ssGSEA scores for reduced gene signatures (containing only genes that have human orthologues) are highly correlated with the ShcA genotype in mice (Supplementary Fig. 12). Spearman's correlations between each signature and expression values of specific genes (*GZMB*, *CD8A* and *PD-L1*) were then computed. For visualization purposes, patients were ranked-ordered and stratified in quartiles, and the mean expression value for each gene and each quartile was computed.

**Statistical analysis.** For all *in vitro* studies, three independent experiments with at least three biological replicates per experimental group were performed, unless mentioned in the figure legends. Data were normalized to the standard or control as appropriate. *In vivo* orthotopic tumour studies in WT and CD8$^{-/-}$ or IFNγ$^{-/-}$ mice were performed with 4–6 age-matched mice (inoculated with tumour in both fourth mammary fat pads; $n = 8$–12 tumours) per group. Power analysis using StatMate software showed a sample size of ten tumours per group provided 80% power to detect a difference between means of 155 mm$^3$ with a significance level of 0.05 (two-tailed) between two groups. Significance testing between two groups with non-normal distribution were done with Wilcoxon's

rank-sum test. This included results from immunohistochemical staining (Figs 1e and 3d, and Supplementary Figs 2a–c, 6a,b and 9a,b), quantitative PCR (Figs 2f,e and 5b,c) and flow cytometry (Fig. 1f,g). Significance testing between two groups assumed to have normal distribution were done using two-sample *t*-test with two-tailed 5% significance level. This includes all *in vitro* studies (Figs 2c,e, 4c and 5a, and Supplementary Figs 5c,d and 7a) and transgenic mice tumour-onset study (Figs 1b and 6d,e). For tumour outgrowth graphs, multiple *t*-test with Holm–Sidak method (Figs 1c,d, 4e and 6a, and Supplementary Fig. 1e,f) without assuming consistent SD was used.

**Data availability.** The TCGA data referenced during the study are available in a public repository from the TCGA Research Network website (http://cancergenome.nih.gov/). The RNAseq data that support the findings of this study have been deposited in the NCBI Sequence Read Archive database under the accession code SRP092760. All the other data supporting the findings of this study are available within the article and its Supplementary Information file, and from the corresponding author upon reasonable request.

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

## Acknowledgements

R.A. dedicates this manuscript in loving memory of Myung Choo Ahn. We thank Dr Peter Siegel for critical reading of the manuscript. This work was supported by CIHR (MOP-133670) and CCSRI (702060) grants to J.U.-S. K.K.M. and A.B. are supported by CIHR (MOP-115000, 137149, 142227). C.L.K. acknowledges grants from the CFI (33902) and NSERC (RGPIN-2016-04911). We further acknowledge support from the small animal research and pathology core facilities at the Lady Davis. We also acknowledge Genome Quebec for the RNAseq studies. J.U.-S. is the recipient of a CIHR New Investigator salary support award. C.L.K. is supported by a Junior I FRSQ salary support award. W.J.M. acknowledges a Canada Research Chair in Molecular Oncology. R.A., Y.K.I. and N.D.J. are supported by FRSQ Doctoral Awards.

## Author contributions

R.A. and J.U.-S. designed the experiments and analysed the data. R.A., V.S., A.M.B., Y.K.Y., Y.K.I., M.C.F. and S.T. performed the experiments. S.H., N.D.J. and C.L.K. analysed the RNAseq data and performed the bioinformatics analyses. T.P., A.E.K., W.J.M. and K.K.M. provided reagents and expertise. R.A., C.L.K. and J.U.-S. wrote and edited the manuscript.

## Additional information

**Competing financial interests:** The authors declare no competing financial interests.

