## [Peer Review File · Nature Communications]

Reviewers' comments:

Reviewer #1 (Remarks to the Author):

In this manuscript, the authors dissect the role of ShcA in a mouse model of mammary carcinoma. They show that phosphorylation of specific tyrosine residues (particularly Y239 and Y240) of ShcA modulates the relative amount of STAT3 and STAT1 tyrosine phosphorylation. This has a major effect on the functional in vivo syngeneic immunogenicity of these tumor cells.

The authors present a wealth of cellular, genetic, and immunologic data to support their hypothesis. Overall, they make a good case that the relative activation states of STAT3 and STAT1 lead to a complex interplay that affects the effective immunogenicity of tumor cells, at least in this one mouse model. The relevance beyond that is not quite as clear. In particular, the following points should be considered:

1. The authors nicely exploit the ability to mutate Y239/Y240 or Y313 to modulate the relative phosphorylation levels of STAT1 and STAT3. However, in the absence of exogenous interferon stimulation, it is not clear that STAT1 activation is routinely found in primary human breast cancers. The authors should address this issue more directly.
2. A related point concerns the mechanism by which mutations in ShcA, particularly of Y313, leads to phosphorylation of STAT1. Is this a miscue of an endogenous physiological signal that is being mediated by ShcA, or does this reflect a more complex pathway (perhaps involving interferons)?
3. The authors leverage gene expression signatures based on the mutations in the tyrosine residues of ShcA. However, these signaling events (and hence these signatures) presumably never occur in a primary human breast cancer. Thus, is there any new information that can be derived from these signatures that would not be derived (with a higher signal-to-noise) from the closely related STAT1 and STAT3 signatures?
4. The authors might try to clarify further the divergent effects of the Y313 mutation on the response to immune checkpoint blockade versus tumor vaccines.

Minor points

5. Some of the speculation in the Discussion perhaps wanders too far afield. For example, it is an oversimplification to call an antibody like trastuzumab a "tyrosine kinase inhibitor" (page 21), as it is not very effective at blocking signals generated from Her2, and in fact activates Her2 initially, likely through cross-linking. Thus, the whole paragraph on Src and trastuzumab resistance should be tempered.
6. The issue of the role of STAT3 activation in immune effector cells is complex, and this event may decrease function of both antigen presenting cells and cytotoxic T cells. Thus, the contention that STAT3 inhibition would only have value in the epithelial (i.e., tumor) cells should perhaps be more balanced.
7. It is not clear what is meant by the first part of the title, "Phospho-tyrosine Signaling Networks Dynamically Control the Balance between STAT1 and STAT3 Activity...." The changes shown between STAT1 and STAT3 do not seem to be dynamic, but are the results seen statically after mutation of specific residues. It is also not clear what network is being referred to. The authors may wish to reconsider the title to more closely hew to the results they present (as is done with the running title).

Reviewer #2 (Remarks to the Author):

The Manuscript by Ahn, Ursini-Siegel and co-workers reports how ShcA, by controlling STAT1 and STAT3 activation, can control the immune response to tumors in the MMTV-PyMT breast cancer model. The data are supported by data from other mouse models and by analysis of human expression data from breast cancer patients. Although the manuscript does not point to a clear path forward to alter ShcA signaling or the balance between STAT1 and STAT3 signaling, the analysis of the effects on the adaptive immune response as well as the signaling processes within cancer cells that controls these responses advances our understanding of how immune suppression/control can be controlled by cancer cells. As such, the findings reported in the manuscript are of high interest to the community.

Unfortunately, it was one of the hardest manuscripts to read that I have reviewed in years: combining a very complicated experimental design, omission of essential background information, and need for extensive, careful editing of the manuscript. Sometimes the authors are even confusing themselves: in the Abstract, the authors state: "Phosphorylation of Y239/Y240 residues on ShcA potently and selectively reduces STAT3 activation". However, it is the inactivating mutation of residue 239/240 that leads to reduction of STAT3 levels, and hence the interpretation of that data would be that the normal function of Y239/240 is to activate STAT3.

I have a substantial amount of comments listed below that hopefully will help the authors improve data presentation and interpretation. These comments should, however, be taken as examples and are not an exhaustive list of where the manuscript needs improvement.

Comments:

1. Example of poor editing: page 5, line 13: "While tumor onset was unaffected (Extended Data Fig 1a-d), the growth potential of two independent MT/Shc2F/2F breast tumors was selectively impaired in mice that contain CD8+ T cells (CD8+/+) or retain intact IFN γ responses (IFN γ +/+), but not in their immunodeficient (CD8-/-, IFN γ -/-) counterparts". This sentence covers results represented by four different figure panels! Furthermore, as a reader it is not clear what the MT/Shc2F/2F tumors are being compared to. Finally, "growth potential" is a very imprecise term.

2. Example of poor editing: The Abstract states "Finally, the ability of breast tumors to CAPITALIZE on STAT1-driven anti-tumor immunity in response to diminished tyrosine kinase signaling can be overcome by a compensatory hyper-activation of STAT3". This makes no sense: a tumor doesn't capitalize on anti-tumor immunity, as that would make the tumor shrink.

3. Lack of discussion: Page 6, second paragraph, first line "In contrast, IFN γ and CXCL9 mRNA levels are specifically increased in MT/Shc2F/2F tumors (Fig. 2f)...". How do the authors square this with the statement in the paragraph just above: "We validated that MT/Shc313F/313F breast cancer cells uniformly and basally upregulated many IFN γ -responsive genes, including CXCL9 as well as components of the antigen processing and presentation machinery (Fig. 2c-e; Extended Data Fig. 4a-c)". The authors completely omit any discussion of why they report in Figure 2c that only the Shc313F cells upregulated CXCL9 protein, while in figure 2f it is only the Shc2F lines that have increased CXCL9 mRNA levels.

4. Lack of quantification of data in western blots: On page 7, last line, the authors state: "While STAT1 loss increased Y705-STAT3 phosphorylation in MT/ShcA+/+ cells, it did not rescue pY705-STAT3 levels in MT/Shc2F/2F cells, suggesting that the Y239/240-ShcA phosphorylation sites directly regulate Y705-STAT3 phosphorylation." There is just one western blot, with no replicates, to support this statement.

5. Misstating what is shown on figure: On p. 6, last line: "Immunoblot analysis shows that STAT1 and pY701-STAT1 levels are basally and uniformly elevated in MT/Shc313F/313F cells." Actually, it is not "uniformly": in Fig. 3a, there is no phospho-STAT3 for the 6737 cell line.

6. The Introduction could be much more to the point – it doesn't provide much background on ShcA, the link between Shc and STAT1/3, or the role of STAT1 and STAT3 in epithelial cells versus immune cells. Instead, the Introduction starts with a long section on breast cancer subtypes. This is largely irrelevant to the present study that really focuses on one model of breast cancer (representing luminal B breast cancer). Along these lines, it doesn't help that the authors later in the introduction state "Amplification, overexpression or tyrosine phosphorylation of ShcA is enriched in HER2+ and basal breast cancers and is an independent predictor of inferior patient outcome." The MMTV-PyMT model is not a model of these subtypes. However, it would have helped if the authors had mentioned that ShcA is activated downstream of PyMT. That is critical information for understanding the manuscript (not all tumor models might be appropriate for looking at ShcA activity).

7. The description of the Statistics mentions only 2-tailed paired t-tests, but not whether ANOVA was performed, or whether the authors performed multiple comparisons corrections, or performed power analysis. The description (and possibly the analysis) needs a bit more statistical depth. The distribution of the values (the authors should be credited for using dot-blots and not bar graphs) strongly suggest that none of the major conclusion will be altered by a more careful analysis, but nevertheless, it is important that the appropriate statistical analysis are performed.

8. The Discussion needs a lot more editing and should be more focused on the presented data and on putting that data in context: What activates ShcA in human breast cancer? Or are the ShcA mutations just a means to manipulate STAT1/STAT3 signaling - or do the findings have implications for both ShcA activation and on other means of activation STAT1/STAT3?

9. In the Results sections, it wasn't clear how MMTV/NIC mice or how the ShcA flx allele were used. That is because the nomenclature for these mice was not the same in the Results and the Methods sections.

10. The description of the generation of the cell lines should be expanded (how long did it take to establish these lines, what is the morphology of the cells in vitro and what is the pathology of the tumors formed from these cell lines in vivo? It wouldn't change the major conclusions of the study if the cell lines don't retain full epithelial morphology, since the study is well controlled. Nevertheless, there should be a description of the cell lines and the tumors derived from them, because most cell lines established from MMTV-PyMT loose epithelial morphology and do not form adenocarcinomas in vivo. This could change the interpretation/implications of the data.

11. Last line p. 6 "STAT1 levels are elevated in 50% of MT/Shc2f/2f cells", should be "cell lines" instead of cells (otherwise gives the impression of heterogeneity between cells of the individual lines).

12. Organization of Fig. 2 is very strange (panels not organized alphabetically). Fig. 3 is also disorganized.

13. Fig. 2a is very strange, 12 genes represent 8 pathways? That seems like over-interpretation of what was found.

14. In Fig. 2f – what is the statistics comparing? Looks like the cell lines were pooled for the analysis but this is not clear.

15. Fig. 3d: the colors in the figure legend and on graph are not the same (pink or peach for Shc2F-SD?).

16. The model figures are confusing rather than helping the reader understand the data.

17. Page 8, lines 4 and 3 from the bottom "Histological assessment reviewed ..." – the authors need to make clear what cell lines they are referring to.
18. The scales on Fig. 4e and f should be the same across all cell lines (for time and volume) so data obtained using the different cell lines can be related to each other.
19. The authors should be careful and not simply equating specific Shc mutations with STAT high or low activity. They don't discuss whether the specific mutation could result in other differences in signaling than STAT activity and how that might contribute to the observed phenotypes.
20. For Fig. 5A, it is not clear how IFN-gamma affects PD-L1 levels as control with and without IFN-gamma are both normalized to "1". As a reader, one would like to see what effect IFN-gamma has on PD-L1 expression on these cell lines.
21. Fig. 6d appears to be bar graphs representing dot plots, so the reader never gets to see the data that resulted in the the R-values mentioned in the text – this is not very satisfying. Do the cited p-values represent p-values of the raw data or of the "bar graphed" data?
22. Fig. S1 should use same scales for all related graphs (on x and y).
23. Fig. S2 is very strange: what did the authors gate on, why are there so few cells in some of the plots in S3d? Is the quantification in the main figures?
24. Fig. S3b – the dark blue and black are almost impossible to differentiate on my print-out. More importantly, why are the cell lines of the same genetic background not grouping together?
25. On Fig. S7C, what is on the y-axis for the dot blots? SSC, FSC, or?
26. On Fig. S8a – why is the top panel stating "ShcA" and the bottom "WT" – are these actually the same cell line - with the additional STAT1 or STAT3 modulation?
27. On page 8, line 7, "STAT1, but not STAT3, increased basal (MT/Shc313F/313F) and IFN γ -inducible (MT/ShcA+/+; MT/Shc2F/2F) surface MHC class I expression on breast cancer cells (Fig. 4c; Extended Figure 6)." This is very confusing, because the experimental design is not clear from the convoluted sentence.
28. The data on Fig. 4e and Fig. 1c/d should be the same (it appear to be essentially the same experiment), but does not appear to be 100% consistent, especially for the ShcA wild type.

Reviewer #3 (Remarks to the Author):

Ahn and colleagues used MMTV/PyVmT (MT) transgenic mouse models carrying specific mutations in the scaffolding protein ShcA (SHC1) to demonstrate the interaction between ShcA, a scaffolding protein involved in tyrosine kinase signal transduction, and STAT1 and STAT3 proteins involved in immune surveillance and immune suppression. The work is a continuation of prior work by the authors and provides new evidence on the interplay of tyrosine kinase signaling and immune response in cancer.

The in vivo experiments with the transgenic mice appear very well designed and well executed, and the conclusions reached are supported by the data presented. The manuscript is well written and the results are presented clearly.

To validate their in vivo findings in human breast cancers, the authors first derived transcriptional signatures specific for the two ShcA genotypes MT/Shc2F/2F and MT/Shc313F/313F relative to wild-type SchA, and “translated” these signatures to human by selecting the signature genes with human orthologs. Finally, they used these signatures to interrogate human breast tumors from the TCGA with respect to the extent to which these genotype signatures were represented in human tumors. In parallel, they developed transcriptional signatures for STAT1 and STAT3 using previously reported gene targets and similarly assessed the extent to which these signatures were overrepresented in human breast tumors.

Although the methodology for carrying out the above analysis is well described, some clarifications on missing details are still required. These are listed below.

1. It was not clear through which comparison the Shc2F-specific and 313F-specific genes were identified. Was it Shc2F vs WT and 313F vs WT, or Shc2F vs rest (313F and WT) and 313F vs rest? Also for the common signature, was it Shc2F and 313F vs WT? Please clarify in the supplementary methods.
2. It would be useful to demonstrate that the “translated” ShcA genotypic signatures (i.e. involving only genes with human orthologs) are highly correlated with the Shc genotype in mice. Perhaps a scatterplot of Shc2F vs 313F enrichment scores showing all cell line replicates using different colors/symbols for each of the three genotypes. This will provide evidence for the validity of the ortholog-based signatures.
3. Similarly with the STAT1, STAT3 signatures. It would be useful to demonstrate that these signatures correlate with the relative protein or phospho-protein levels measured in the different mouse models. Also, since protein and phosphoprotein expression measurements (RPPA) are available in the TCGA, it would be informative to show how these transcriptional signatures correlate with mRNA, protein and phosphoprotein levels of STAT1 and STAT3 in the human breast tumors. Additional evidence would be needed show that these signatures indeed track STAT1 and STAT3 related pathway activities.
4. It is not clear whether the authors developed their own implementation of ssGSEA or used existing software available by the Broad Institute group who developed the method or other software packages available (e.g. in Bioconductor). They should clarify and provide appropriate reference if an existing software implementation was used.

Reviewer #4 (Remarks to the Author):

Ahn R et al. demonstrated an important and clinically relevant role of phospho-tyrosine ShcA signaling networks involved in STAT1/STAT3 activation, regulating the balance between immunostimulatory and immunosuppressive tumor microenvironment in breast cancers (BC), both in mice and humans. The amount of work is impressive and the demonstration is quite rigorously performed in the MMTV/PyvMT mouse tumor models, and then, somewhat confirmed in the TCGA BC data base. I have mainly minor comments :

Fig. 1E : the bar scale of IHC should be shown

Fig. 1F-G : the raw data (instead of the fold changes) must be depicted.

Fig. 6B : does the vaccine efficacy depend on the MHC class I expression levels of the ip immunizing cells ? results of the immunizing capacities of the STAT1- or STAT3 –CRISPR cell counterparts ?

Fig. 6E : comment on the percentages (and n absolute numbers) of human BC primary tumors falling into each categories in the TCGA data base ?

Reviewer #1:

1. The authors nicely exploit the ability to mutate Y239/Y240 or Y313 to modulate the relative phosphorylation levels of STAT1 and STAT3. However, in the absence of exogenous interferon stimulation, it is not clear that STAT1 activation is routinely found in primary human breast cancers. The authors should address this issue more directly. A related point concerns the mechanism by which mutations in ShcA, particularly of Y313, leads to phosphorylation of STAT1. Is this a miscue of an endogenous physiological signal that is being mediated by ShcA, or does this reflect a more complex pathway (perhaps involving interferons)?

The reviewer brings up an excellent point. To test this, we measured expression levels of the IFN family of cytokines produced for our cell lines, both at the mRNA and protein levels. We now show that many IFN family members are either absent or expressed at exceedingly low levels. Further analysis of the RNAseq data generated from four independent cell lines per genotype (MT/ShcA^{+/+}, MT/Shc^{2F/2F} and MT/Shc^{313F/313F}) show that most IFN genes (IFN α 1, IFN α 2, IFN α 5, IFN α 6, IFN α 7, IFN α 9, IFN α 14, IFN γ) are undetectable (based on the absence of detectable reads). For two IFN genes, (IFN α 4 and IFN β 1) we could detect **exceedingly low** levels of expression among the individual cell lines (range: 1-15 reads), which falls well below the cut off of 100 reads that we used to identify differentially expressed genes (see Supplementary Methods, page 4 for description of our stratification criteria). This data is now shown in **Table S2** of the revised manuscript. To further convince ourselves that differential expression of the various IFN proteins do not contribute to the increased STAT1 activation observed in MT/Shc^{313F/313F} cells, we also performed ELISA assays on four independent cell lines per genotype (MT/ShcA^{+/+}, MT/Shc^{2F/2F} and MT/Shc^{313F/313F}). We show that both IFN α 1 and IFN γ protein levels were undetectable from any cell line tested. Moreover, IFN β 1 levels are exceedingly low and comparable among all cell lines tested. This data is now shown as **Reviewer Figure 1** and includes the standard curves for each assay to show that the absence of appreciable IFN levels is not due to technical limitations. Combined, this data suggests that the absence of pY313-ShcA signaling does not augment STAT1 signaling by establishing an autocrine IFN signaling loop. Instead, these data suggest pY-ShcA signaling basally represses intracellular machinery that permits endogenous STAT1

signaling in the absence of external IFN-driven stimuli. This is now described in the revised manuscript (page 7; lines 21-24 and page 8; lines 1-2).

2. ***The authors leverage gene expression signatures based on the mutations in the tyrosine residues of ShcA. However, these signaling events (and hence these signatures) presumably never occur in a primary human breast cancer. Thus, is there any new information that can be derived from these signatures that would not be derived (with a higher signal-to-noise) from the closely related STAT1 and STAT3 signatures?***

The reviewer is correct that these mutations in ShcA (Y239/240F and Y313F) are not likely to be found in human breast cancer. Instead, we use them as a surrogate to genetically interrogate the mechanistic basis for how specific perturbation of signaling pathways downstream of each individual phospho-site affects immune suppression.

ShcA signaling in human breast cancer: Significant experimental evidence supports the fact that ShcA expression and activity vary widely among individual breast cancers. (1) Most receptor and cytoplasmic tyrosine kinases that are clinically relevant to human breast cancer (EGFR, ErbB2, ErbB3, Met, FGFR, INSR, IGFR, etc.) recruit ShcA and require ShcA to transduce their oncogenic signals (see Reviewer Table 1). (2) ShcA protein levels vary widely across individual breast cancers (as assessed by TMA staining of invasive ductal carcinomas). Importantly, elevated ShcA protein levels are enriched in HER2 and basal breast tumors and associated with inferior clinical outcome (PMID: 20924104). (3) This is consistent with the fact that ShcA resides within a newly identified amplicon (Chr1q21-23) that is observed in roughly 15% of all breast cancers (PMID: **26109346**). Importantly ShcA gene amplification, is enriched in basal and luminal/p53 negative breast cancers, which is consistent with our immunohistochemical studies (see point 2). (4) Previous studies have shown that increased ShcA tyrosine phosphorylation in primary breast cancers correlates with lymph node status, tumor stage and recurrence (PMID: 1458473). The absence of high-quality commercially-available phospho-ShcA antibodies (specific for the Y239/240 and Y313 phospho-sites and that are amenable to immunohistochemical staining of paraffin-embedded sections) precludes us from extended these studies to individual ShcA tyrosine phosphorylation sites (unpublished observations). Combined, these studies provide a solid rationale for understanding the molecular basis by which the ShcA adaptor transduces oncogenic signals that promote breast cancer progression.

Relevance of ShcA mutants employed in this study to human breast cancer immune suppression: Our genetic approach to manipulating ShcA signaling in breast tumors model: (1) tumors that intrinsically possess reduced tyrosine kinase/ShcA signaling or that (2) inducibly repress the ShcA signaling axis in response to tyrosine kinase inhibitors.

Intrinsic Immune Suppression: MT/ShcA^{+/+} cells employed herein model breast tumors that possess an activated tyrosine kinase/ShcA axis, which simultaneously activates STAT3 and represses STAT1 in breast tumors to establish immune suppression.

Immune Surveillance: MT/ShcA^{2F/2F} breast tumors employed herein represent those human breast tumors that are low in tyrosine kinase/ShcA signaling (intrinsic or in response to tyrosine kinase inhibitors). This genetic tool provides new knowledge that suppression of STAT3 signaling downstream of ShcA is sufficient to restore immune surveillance.

Acquired Immune Suppression: MT/ShcA^{313F/313F} cells represent tumors that are debilitated in tyrosine kinase/ShcA signaling (intrinsic or in response to TKIs) but that re-activated downstream signaling pathways (both STAT3 dependent and independent) to re-acquire immune suppression. Indeed, the observation that STAT3 loss impaired, but did not ablate, MT/ShcA^{313F/313F} tumor growth, specifically in an immunocompetent background, supports the presence of additional STAT3-independent mechanisms in acquired immunosuppression (Figure 4). However, identification of these pathways are beyond the scope of this manuscript.

Relevance of ShcA mutant gene signatures: We did not mean to imply that the Shc2F and Shc313F signatures are surrogates for STAT1 and STAT3 transcriptional activity. This would be an oversimplification as the ShcA signaling axis recruits numerous downstream signaling molecules that affect many other biological processes (cell proliferation, survival, migration, invasion, angiogenesis) beyond immune suppression (PMID: 18604176; 22970934). Moreover, we also recognize that ShcA-independent mechanisms also likely control STAT1 and STAT3 activity. Finally, these ShcA-regulated gene signatures were derived from breast cancer cells in vitro, and thus represent direct ShcA-driven gene expression changes. We recognize that these signatures are being used to interrogate whole tumor datasets, but the fact that they were originally derived from breast cancer cells increases the likelihood that they represent transcriptional responses that are enriched in the epithelial compartment of breast tumors. While the STAT1 and STAT3 signatures employed herein identified likely important roles for these transcription factors in controlling tumor immunity, we cannot discriminate whether the STAT-regulated genes are derived from breast tumor cells or stromal cells. Thus, we believe that these ShcA mutant, STAT1 and STAT3 gene signatures will provide complementary and useful research tools to the scientific community to begin to decipher how tyrosine kinase/ShcA-driven transcriptional responses globally contribute to breast cancer development, including immune suppression.

3. *The authors might try to clarify further the divergent effects of the Y313 mutation on the response to immune checkpoint blockade versus tumor vaccines.*

We have added additional text to the discussion to further clarify this statement. It reads as follows (see page 19, lines 19-24; page 20 lines, 1-6 of the revised manuscript): Unique loss of pY313-ShcA signaling, through re-activation of STAT3-dependent and -independent immunosuppressive signals renders tumors insensitive to immune checkpoint inhibitors. We propose that the strong immunosuppressive state imposed by Shc313F-like breast tumors (STAT1^{High}/STAT3^{High}) limits the therapeutic potential of strategies that aim to block T cell peripheral tolerance, such as PD1 immune checkpoint blockade. Indeed, we could identify human breast tumors that coordinately increase STAT1- and STAT3-driven transcriptional responses in human breast cancers, highly the relevance of this patient cohort (Figure 6). We provide the first experimental evidence that the elevated STAT1 response in these tumors could sensitize them to immunotherapeutic approaches that serve to augment STAT1-driven anti-tumor immune responses, including vaccine-based strategies. These studies demonstrate that the STAT1/STAT3 ratio may represent a useful biomarker to not only predict the degree of breast cancer immune suppression but also the type of immunotherapy that is most likely to yield durable clinical responses in individual breast tumors.

4. Some of the speculation in the Discussion perhaps wanders too far afield. For example, it is an oversimplification to call an antibody like trastuzumab a “tyrosine kinase inhibitor” (page 21), as it is not very effective at blocking signals generated from Her2, and in fact activates Her2 initially, likely through cross-linking. Thus, the whole paragraph on Src and trastuzumab resistance should be tempered.

We agree that the connection between Src hyperactivation in Trastuzumab resistant breast tumors and STAT3 re-activation may be too speculative at this point without additional experimental evidence. We removed the paragraph in question from the revised discussion.

5. The issue of the role of STAT3 activation in immune effector cells is complex, and this event may decrease function of both antigen presenting cells and cytotoxic T cells. Thus, the contention that STAT3 inhibition would only have value in the epithelial (i.e., tumor) cells should perhaps be more balanced.

We agree with the reviewer and have included significant text into the discussion to present a more balanced role for STAT3 in the immune microenvironment. The addition text can be found on page 20, lines 16-24; and page 21, lines 1-5 of the revised manuscript and is as follows: Inhibition of tumor intrinsic STAT3 and in numerous cell types within the immune microenvironment has the potential to significantly impede tumor immune suppression. For example, inhibition of STAT3 signaling in immune cell types with anti-tumorigenic properties, including NK cells and cytotoxic T lymphocytes, significantly increases their tumoricidal properties in numerous cancer types. Moreover, increased STAT3 signaling in dendritic cells inhibits their maturation and subsequent ability to serve as antigen presenting cells in educating anti-tumor immune responses. However, other studies suggest that STAT3 signalling in immune cells is also important for the initiation of inflammatory responses, which are essential to initiate and educate anti-tumor T cell responses. Moreover, in regulatory T cells, STAT3 is engaged downstream of inflammatory cytokine signaling, such as the IL6 receptor, to limit their immunosuppressive properties. The current study suggests that inhibition of STAT3 signaling, specifically in the tumor epithelium, robustly engages immune surveillance. Our observations suggest that the development of inhibitors that bind the Y239/240-ShcA phospho-sites may represent an alternative therapeutic strategy to inhibit STAT3 activation in breast cancer cells.

6. It is not clear what is meant by the first part of the title, “Phospho-tyrosine Signaling Networks Dynamically Control the Balance between STAT1 and STAT3 Activity....” The changes shown between STAT1 and STAT3 do not seem to be dynamic, but are the results seen statically after mutation of specific residues. It is also not clear what network is being referred to. The authors may wish to reconsider the title to more closely hew to the results they present (as is done with the running title).

We appreciate the reviewer's suggestion and have taken it under advisement. We have now modified the title to *“The Shc1 Scaffold protein simultaneously balances Stat1 and Stat3 activity in breast cancers to promote immune suppression and resistance to immunotherapy.”*

Reviewer #2:

1. **Sometimes the authors are even confusing themselves: in the Abstract, the authors state: “Phosphorylation of Y239/Y240 residues on ShcA potently and selectively reduces STAT3 activation”. However, it is the inactivating mutation of residue 239/240 that leads to reduction of STAT3 levels, and hence the interpretation of that data would be that the normal function of Y239/240 is to activate STAT3.**

We corrected this error and have thoroughly read through the manuscript to rectify any other issue. All changes made to the text are in blue font.

2. **Example of poor editing: page 5, line 13: “While tumor onset was unaffected (Extended Data Fig 1a-d), the growth potential of two independent MT/Shc2F/2F breast tumors was selectively impaired in mice that contain CD8+ T cells (CD8^{+/+}) or retain intact IFN γ responses (IFN γ ^{+/+}), but not in their immunodeficient (CD8^{-/-}, IFN γ ^{-/-}) counterparts”. This sentence covers results represented by four different figure panels! Furthermore, as a reader it is not clear what the MT/Shc2F/2F tumors are being compared to. Finally, “growth potential” is a very imprecise term.**

This sentence has been re-worded and now reads as follows: Tumor onset of two independent MT/ShcA^{+/+}, MT/Shc^{2F/2F} and MT/Shc^{313F/313F} cell lines were unaffected in CD8^{+/+} versus CD8^{-/-} (Supplementary Fig 1a, b) or IFN γ ^{+/+} versus IFN γ ^{-/-} animals (Supplementary Fig 1c, d). In contrast, the growth rate of two independent MT/Shc^{2F/2F} breast tumors was selectively impaired in mice that contain CD8⁺ T cells (CD8^{+/+}) or retain intact IFN γ responses (IFN γ ^{+/+}) relative to their immune-deficient (CD8^{-/-}, IFN γ ^{-/-}) counterparts.

3. **Example of poor editing: The Abstract states “Finally, the ability of breast tumors to CAPITALIZE on STAT1-driven anti-tumor immunity in response to diminished tyrosine kinase signaling can be overcome by a compensatory hyper-activation of STAT3”. This makes no sense: a tumor doesn’t capitalize on anti-tumor immunity, as that would make the tumor shrink.**

We have rephrased the sentence in the abstract. It now reads as follows: Finally, the ability of diminished tyrosine kinase signaling to initiate STAT1-driven immune surveillance in breast tumors can be overcome by a compensatory hyper-activation of STAT3.

4. **Lack of discussion: Page 6, second paragraph, first line “In contrast, IFN γ and CXCL9 mRNA levels are specifically increased in MT/Shc2F/2F tumors (Fig. 2f)...”. How do the authors square this with the statement in the paragraph just above: “We validated that MT/Shc313F/313F breast cancer cells uniformly and basally upregulated many IFN γ -responsive genes, including CXCL9 as well as components of the antigen processing and presentation machinery (Fig. 2c-e; Extended Data Fig. 4a-c)”. The authors completely omit any discussion of why they report in Figure 2c that only the Shc313F cells upregulated CXCL9 protein, while in figure 2f it is only the Shc2F lines that have increased CXCL9 mRNA levels.**

We have modified the text to clarify this point. Our data suggest that tyrosine kinases engage pY313-ShcA to restrain IFN-regulated responses but that re-activation of IFN-driven transcriptional responses in MT/Shc^{313F/313F} tumors is not sufficient to overcome immune suppression *in vivo* (Fig. 1c, d). In contrast, inhibition of pY239/240ShcA-coupled signaling pathways in mammary tumors (MT/Shc^{2F/2F}) specifically elicits immune surveillance responses *in vivo* (Fig. 1c, d; Supplementary Fig. 1), despite the fact IFN γ -driven signaling responses are inhibited in these cell lines *in vitro* (Fig. 2). We show that *IFN γ* and *CXCL9* mRNA levels are specifically increased in MT/Shc^{2F/2F} tumors (Fig. 2h), suggesting that loss of pY239/240ShcA-dependent signaling in breast cancer cells is sufficient to prime mammary tumors to IFN γ -dependent activation of immune surveillance pathways *in vivo*. This text is found on page 7, lines 3-11 of the revised manuscript.

5. ***Lack of quantification of data in western blots: On page 7, last line, the authors state: “While STAT1 loss increased Y705-STAT3 phosphorylation in MT/ShcA+/+ cells, it did not rescue pY705-STAT3 levels in MT/Shc2F/2F cells, suggesting that the Y239/240-ShcA phosphorylation sites directly regulate Y705-STAT3 phosphorylation.” There is just one western blot, with no replicates, to support this statement.***

We apologize for this oversight. These immunoblots are representative of triplicate experiments and quantification of the three immunoblots is now shown in Supplementary Figure 7 (panels a and b).

6. ***Misstating what is shown on figure: On p. 6, last line: “Immunoblot analysis shows that STAT1 and pY701-STAT1 levels are basally and uniformly elevated in MT/Shc313F/313F cells.” Actually, it is not “uniformly”: in Fig. 3a, there is no phospho-STAT3 for the 6737 cell line.***

This sentence has been modified and now reads as follows (page 7, lines 14-16 of the revised manuscript): Immunoblot analysis shows that STAT1 levels are basally elevated in all MT/Shc^{313F/313F} cells, coinciding with increased Y701-STAT1 phosphorylation in the majority of them.

7. ***The Introduction could be much more to the point – it doesn’t provide much background on ShcA, the link between Shc and STAT1/3, or the role of STAT1 and STAT3 in epithelial cells versus immune cells. Instead, the Introduction starts with a long section on breast cancer subtypes. This is largely irrelevant to the present study that really focuses on one model of breast cancer (representing luminal B breast cancer). Along these lines, it doesn’t help that the authors later in the introduction state “Amplification, overexpression or tyrosine phosphorylation of ShcA is enriched in HER2+ and basal breast cancers and is an independent predictor of inferior patient outcome.” The MMTV-PyMT model is not a model of these subtypes. However, it would have helped if the authors had mentioned that ShcA is activated downstream of PyMT. That is critical information for understanding the manuscript (not all tumor models might be appropriate for looking at ShcA activity).***

We have removed the paragraph relating to breast cancer subtypes from the introduction and put the focus on the importance of immune surveillance in controlling cancer development and sensitivity to immunotherapies. As suggested, we also

included a paragraph in the introduction that specifically describes the importance of the ShcA signaling pathway in tyrosine kinase signaling (page 4; lines 5-16). It reads as follows: In this regard, tyrosine kinases rely on a core set of signaling intermediates to transduce oncogenic signals. One such scaffolding protein, called Shc1 (or ShcA), is recruited to multiple tyrosine kinases and is essential for tumor initiation, progression and metastatic spread in breast cancer mouse models. The mammalian *ShcA* gene encodes three proteins that are generated through differential promoter usage (p66) or alternative translational start sites (p46, p52). While p66ShcA transduces oxidative stress responses, p46/52ShcA employ numerous domains and motifs to transduce phospho-tyrosine dependent signals downstream of receptor and non-receptor tyrosine kinases. ShcA translocates from the cytosol to the plasma membrane where it interacts with phospho-tyrosine residues in activated tyrosine kinases. These interactions are mediated by either the PTB or SH2 domains of ShcA. In turn, tyrosine kinases phosphorylate three tyrosine residues (Y239/Y240 and Y317- analogous to Y313 in mice) within the central collagen homology 1 (CH1) domain of ShcA. Once phosphorylated, these tyrosines serve as docking sites for other PTB- and SH2-containing proteins to activate diverse signaling pathways, including but not limited to the Ras/MAPK and PI3K/AKT pathways.

The reviewer also asked us to mention that ShcA signaling is important for Polyoma middle T Virus induced transformation. This is an excellent point and has now been incorporated (page 5, lines 7-10). The text reads as follows: We chose the Polyomavirus middle T antigen (MT) mouse model for two reasons. First, MT-induced transformation recapitulates all the stages of breast cancer progression in transgenic mice. Second, mutation of the ShcA binding site in MT is sufficient to delay tumor onset and reduce tumor burden in transgenic mice, demonstrating that ShcA-coupled signal transduction is important for MT-induced breast cancer development.

Finally, the reviewer asks that we discuss the link between ShcA and STAT1/3 along with the role of STAT1 and STAT3 in epithelial cells versus immune cells. We have added text to discuss how ShcA is activated in human breast cancers (page 18; lines 9-21). The text reads as follows: Significant experimental evidence supports the fact that ShcA expression and activity varies widely among individual breast cancers and is relevant to patient outcome. For example, ShcA represents a key convergent point downstream of many tyrosine kinases that are functionally important for breast cancer development, including but not limited to EGFR, ErbB2, and MET, FGFR, IGFR1 and Src. Moreover, immunohistochemical staining of invasive breast carcinomas demonstrated that ShcA protein levels vary widely across individual breast cancers, are enriched in the HER2 and basal subtypes and associate with inferior clinical outcome. (3) This is consistent with the fact that ShcA resides within a newly identified amplicon (Chr1q21-23) observed in roughly 15% of all breast cancers. This amplicon is enriched in basal and luminal/p53 negative breast cancers. Previous studies have also shown that increased ShcA tyrosine phosphorylation in primary breast cancers correlates with lymph node status, tumor stage and recurrence. Combined, these studies provide a solid rationale for understanding the molecular basis by which the ShcA adaptor transduces oncogenic signals that promote breast cancer immune suppression.

The previous discussion focused on the role of STAT1 and STAT3 in the mammary epithelial compartment to control breast cancer development and tumor immunity. We have extended this section to include an overview of the role of STAT3 signaling in immune cells in controlling immune suppression. The incorporated text is found on page 20, lines 16-24 and page 21, lines 1-5 of the revised manuscript and reads as follows: Inhibition of tumor intrinsic STAT3 and in numerous cell types within the immune microenvironment has the potential to significantly impede tumor immune suppression. For example, inhibition of STAT3 signaling in immune cell types with anti-tumorigenic properties, including NK cells and cytotoxic T lymphocytes, significantly increases their tumoricidal properties in numerous cancer types. Moreover, increased STAT3 signaling in dendritic cells inhibits their maturation and subsequent ability to serve as antigen presenting cells in educating anti-tumor immune responses. However, other studies suggest that STAT3 signalling in immune cells is also important for the initiation of inflammatory responses, which are essential to initiate and educate anti-tumor T cell responses. Moreover, in regulatory T cells, STAT3 is engaged downstream of inflammatory cytokine signaling, such as the IL6 receptor, to limit their immunosuppressive properties. The current study suggests that inhibition of STAT3 signaling, specifically in the tumor epithelium, robustly engages immune surveillance. We suggest that the development of inhibitors that bind the Y239/240-ShcA phospho-sites may represent an alternative therapeutic strategy to inhibit STAT3 activation in breast cancer cells.

8. *The description of the Statistics mentions only 2-tailed paired t-tests, but not whether ANOVA was performed, or whether the authors performed multiple comparisons corrections, or performed power analysis. The description (and possibly the analysis) needs a bit more statistical depth. The distribution of the values (the authors should be credited for using dot-blots and not bar graphs) strongly suggest that none of the major conclusion will be altered by a more careful analysis, but nevertheless, it is important that the appropriate statistical analysis are performed.*

To address this comment, we have added the following text to the Statistical Analysis under the Methods section: For all *in vitro* studies, three independent experiments with at least three biological replicates per experimental group were performed unless mentioned in the figure legends. Data were normalized to the standard or control as appropriate. In vivo orthotopic tumor studies in WT and CD8^{-/-} or IFN γ ^{-/-} mice were performed with 4-6 age matched mice (inoculated with tumor in both fourth mammary fat pads; n=8-12 tumors) per group. Power analysis using StatMate software showed a sample size of 10 tumors per group provided 80% power to detect a difference between means of 155mm³ with a significance level of 0.05 (two-tailed) between two groups. Significance testing between two groups with non-normal distribution were done with Wilcoxon rank-sum test. This included results from immunohistochemical staining (Fig. 1e, 3d; Supplementary Fig. 2a-c, 6a, 6b, 9a, 9b), qPCR (Fig. 2f, 2e, 5b, 5c) and flow cytometry (Fig. 1f, 1g). Significance testing between two groups assumed to have normal distribution were done using two-sample t test with two-tailed 5% significance level. This includes all *in vitro* studies (Fig. 2c, 2e, 4c, 5a; Supplementary Fig. 5c, 5d, 7a), transgenic mice tumor onset study (Fig. 1b) and Fig. 6d and 6e. For tumor outgrowth graphs, multiple t test with Holm-Sidak method (Fig. 1c, 1d, 4e, 6a;

Supplementary Fig. 1e, 1f) without assuming consistent SD was used. All statistical analyses were performed using GraphPad.

9. ***The Discussion needs a lot more editing and should be more focused on the presented data and on putting that data in context: What activates ShcA in human breast cancer? Or are the ShcA mutations just a means to manipulate STAT1/STAT3 signaling - or do the findings have implications for both ShcA activation and on other means of activation STAT1/STAT3?***

This text was added to the discussion section. We added two pages to the discussion to address this point (see pages 18-19 of the revised manuscript).

10. ***In the Results sections, it wasn't clear how MMTV/NIC mice or how the ShcA flx allele were used. That is because the nomenclature for these mice was not the same in the Results and the Methods sections.***

The nomenclature has been standardized in the results and methods sections. The modified text in the results section (page 6; lines 23-24 and page 7, lines 1-3) now reads as follows: We extended these observations to the MMTV/NIC transgenic mouse model, which expresses an oncogenic ErbB2 variant, coupled to the Cre recombinase, in the mammary epithelium. Bigenic NIC/ShcA^{flx} mammary tumors, which lack ShcA in the epithelial compartment, also upregulate several components of the APP machinery (Supplementary Fig. 4d).

11. ***The description of the generation of the cell lines should be expanded (how long did it take to establish these lines, what is the morphology of the cells in vitro and what is the pathology of the tumors formed from these cell lines in vivo? It wouldn't change the major conclusions of the study if the cell lines don't retain full epithelial morphology, since the study is well controlled. Nevertheless, there should be a description of the cell lines and the tumors derived from them, because most cell lines established from MMTV-PyMT loose epithelial morphology and do not form adenocarcinomas in vivo. This could change the interpretation/implications of the data.***

To address this point, we have characterized the morphology of the respective cell lines in vitro in addition to the histology of their corresponding mammary tumors in vivo. We now show that the various breast cancer cell lines possess variable patterns of cell shape and tumor histology but these changes do not correlate whether individual breast cancer cell lines exhibit immune surveillance or immune suppression phenotypes. This is now incorporated into the results section (page 6; lines 8-10) and is now Supplementary Figure 3 of the revised manuscript.

12. ***Last line p. 6 "STAT1 levels are elevated in 50% of MT/Shc2f/2f cells", should be "cell lines" instead of cells (otherwise gives the impression of heterogeneity between cells of the individual lines).***

This has been corrected.

13. ***Organization of Fig. 2 is very strange (panels not organized alphabetically). Fig. 3 is also disorganized.***

We have re-configured the layout of Figure 2 as requested by the reviewer. For Figure 3, we were constrained by the multiple panels and size of the data elements for each

individual panel. We feel that the current organization of Figure 3 does not compromise the ability of the reader to follow the text and respectfully hope that the reviewer agrees.

14. ***Fig. 2a is very strange, 12 genes represent 8 pathways? That seems like over-interpretation of what was found.***

We used DAVID Functional Annotation Bioinformatics Microarray Analysis platform to systematically analyse and assign gene ontology terms by biological processes (GOTERM_BP_DIRECT) and by molecular function (GOTERM_MF_DIRECT) (PMID: 19033363; 19131956) to the 12 genes. Indeed, they are divided into 8 groups: Metabolism, Signaling, Immune, ECM, Neuronal, Translation, Cytoskeleton, Unassigned (See Reviewer Table 2).

15. ***In Fig. 2f – what is the statistics comparing? Looks like the cell lines were pooled for the analysis but this is not clear.***

We have reanalyzed the data using Mann-Whitney U test to compare the ShcA wild-types as a group to 2F mutants as a group. This was further clarified in the materials and methods section.

16. ***Fig. 3d: the colors in the figure legend and on graph are not the same (pink or peach for Shc2F-SD?).***

This has been corrected.

17. ***The model figures are confusing rather than helping the reader understand the data.***

We have addressed this by adding an overall schematic figure of our proposed model (now Figure 7). We have left the individual model figure elements in Figures 4 and 6 as we feel they are necessary to help the reader grasp the main points of our story. We hope that the overall model figure is helpful and the reviewer agrees with the proposed changes.

18. ***Page 8, lines 4 and 3 from the bottom “Histological assessment reviewed ...” – the authors need to make clear what cell lines they are referring to.***

This has been corrected.

19. ***The scales on Fig. 4e and f should be the same across all cell lines (for time and volume) so data obtained using the different cell lines can be related to each other.***

This has been corrected.

20. ***The authors should be careful and not simply equating specific Shc mutations with STAT high or low activity. They don’t discuss whether the specific mutation could result in other differences in signaling than STAT activity and how that might contribute to the observed phenotypes.***

We have inserted a sentence in the discussion to clarify this point (page 18, lines 5-7 of the revised manuscript).

21. **For Fig. 5A, it is not clear how IFN-gamma affects PD-L1 levels as control with and without IFN-gamma are both normalized to “1”. As a reader, one would like to see what effect IFN-gamma has on PD-L1 expression on these cell lines.**

We have re-plotted the data in Figure 5A generated with or without IFN-gamma treatment on the same X axis to reflect the difference.

22. **Fig. 6d appears to be bar graphs representing dot plots, so the reader never gets to see the data that resulted in the R-values mentioned in the text – this is not very satisfying. Do the cited p-values represent p-values of the raw data or of the “bar graphed” data?**

It is never easy to visualize big datasets like the ones we used in this work, and to convey correlations in a way that is easily understandable by a wide range of readers. Barplots are a very familiar representation for many readers, and that was the main reason for choosing them. P-values were, of course, computed on the raw data.

23. **Fig. S1 should use same scales for all related graphs (on x and y).**

The figure has been modified to reflect the reviewer’s comments.

24. **Fig. S2 is very strange: what did the authors gate on, why are there so few cells in some of the plots in S3d? Is the quantification in the main figures?**

The quantification of the Figure S2 is reflected in Figure 1f and 1g (middle panel).

The CD8⁺/CD69⁺ populations were relatively abundant in the spleens of these mice while they were scarce in breast tumor tissues, which are expected.

25. **Fig. S3b – the dark blue and black are almost impossible to differentiate on my print-out. More importantly, why are the cell lines of the same genetic background not grouping together?**

Our data demonstrates that perturbations in ShcA signaling do not induce significant global gene expression changes under steady state culture conditions (now Figure S4a). Rather, this data suggests that the ShcA adaptor controls a discrete set of genes downstream of activated tyrosine kinases. This is consistent with the fact that ShcA represents one signaling output downstream of tyrosine kinases. With respect to Figure S4b (previously S3b in the original manuscript), in fact, all the MT/Shc^{+/+} tumors cluster together very faithfully. We agree that there is more variability for the MT/Shc^{313F/313F} and MT/Shc^{2F/2F} cell lines. We ask the reviewer to keep in mind that the PCA analysis in Figure S4b represents the entire transcriptome. Given that MT/Shc^{313F/313F} and MT/Shc^{2F/2F} tumor onset is significantly delayed, relative to MT/Shc^{+/+} controls in transgenic mice (see Figure 1), we believe that part of the variability observed in the PCA analysis is because of compensatory ShcA-independent gene expression changes that facilitate eventual mammary tumor development. It is for this reason that we employed very stringent criteria (2 fold differentially expressed; FDR<0.05; >100 reads across ALL four cell lines per genotype) to identify true ShcA-regulated target genes.

26. **On Fig. S7C, what is on the y-axis for the dot blots? SSC, FSC, or?**

The axis is now correctly labelled as FSC-A and is reflected in the figure.

27. ***On Fig. S8a – why is the top panel stating “ShcA” and the bottom “WT” – are these actually the same cell line - with the additional STAT1 or STAT3 modulation?***

This has been corrected in the figure.

28. ***On page 8, line 7, “STAT1, but not STAT3, increased basal (MT/Shc313F/313F) and IFN γ -inducible (MT/ShcA $^{+/+}$; MT/Shc2F/2F) surface MHC class I expression on breast cancer cells (Fig. 4c; Extended Figure 6).” This is very confusing, because the experimental design is not clear from the convoluted sentence.***

We have clarified and elaborated on the experimental design in the manuscript. The revised text is now found on page 9, lines 3-7 and is as follows: Next, we asked how STAT1 and STAT3 regulate basal and IFN γ -induced antigen presentation in our various control and ShcA mutant cell lines. Thus, we assessed the level of surface MHC class I level in MT/ShcA $^{+/+}$, MT/Shc $^{2F/2F}$ and MT/Shc $^{313F/313F}$ cell lines (control, STAT1-deficient or STAT3-deficient) via flow cytometry.

29. ***The data on Fig. 4e and Fig. 1c/d should be the same (it appear to be essentially the same experiment), but does not appear to be 100% consistent, especially for the ShcA wild type.***

Respectfully speaking, we do not agree that the growth patterns observed for the ShcA-WT tumors are appreciably different. We ask the reviewer to keep in mind that these experiments were performed on separate days and minor changes in tumor inoculum for example may account for some of the minor changes observed. Our decision to consistently inject the same cancer cell lines in CD8 $^{+/+}$ versus CD8 $^{-/-}$ animals, serves as the best internal control between different experiments.

Reviewer #3:

1. ***It was not clear through which comparison the Shc2F-specific and 313F-specific genes were identified. Was it Shc2F vs WT and 313F vs WT, or Shc2F vs rest (313F and WT) and 313F vs rest? Also for the common signature, was it Shc2F and 313F vs WT? Please clarify in the supplementary methods.***

To acquire these signatures, we first compared genes that are differentially expressed between the following groups: (1) ShcA-WT versus Shc2F and (2) ShcA-WT versus Shc313F. We then compared these both lists of differentially expressed genes to identify: (a) genes that are commonly differentially expressed in all Shc2F cell lines relative to the rest, (b) genes that are commonly differentially expressed in all Shc313F cell lines relative to the rest (Shc313F-like) and (c) genes that are commonly differentially expressed in both Shc2F and Shc313F cells relative to ShcA-WT cells. This information has now been added to the Supplementary Methods section (page 4).

2. ***It would be useful to demonstrate that the “translated” ShcA genotypic signatures (i.e. involving only genes with human orthologs) are highly correlated with the Shc genotype in mice. Perhaps a scatterplot of Shc2F vs 313F enrichment scores showing all cell line replicates using different colors/symbols for each of***

the three genotypes. This will provide evidence for the validity of the ortholog-based signatures.

We agree with the reviewer this control provides useful information and evidence of the validity of our signature scores, and thus we included the requested plot as Supplementary Figure 11 (described on page 13, lines 9-12 of the revised manuscript). The ssGSEA score computed with the reduced list of genes that have human orthologs indeed correlates, as expected, with the Shc genotype in mice.

3. Similarly with the STAT1, STAT3 signatures. It would be useful to demonstrate that these signatures correlate with the relative protein or phospho-protein levels measured in the different mouse models. Also, since protein and phosphoprotein expression measurements (RPPA) are available in the TCGA, it would be informative to show how these transcriptional signatures correlate with mRNA, protein and phosphoprotein levels of STAT1 and STAT3 in the human breast tumors. Additional evidence would be needed show that these signatures indeed track STAT1 and STAT3 related pathway activities.

We would like to thank the reviewer for suggesting these analyses, which we found to be a clever way of further validating our signatures. These requested analyses are all included in the revised version of the manuscript.

The first control suggested (correlation of STAT1 and STAT3 signatures with the relative protein levels and phospho-protein levels in our mouse models) is now presented as Supplementary Figure 9. We observe a positive correlation of STAT1 and pSTAT1 protein levels with STAT1 signature activation (as computed by ssGSEA scores) in all cases. We also observe a positive correlation between pSTAT3 protein levels and STAT3 ssGSEA scores across all cell lines. Consistent with the function of these transcription factors, the correlation is stronger for STAT1 (spearman correlations for the different features assayed: 0.67-079), than for STAT3. Indeed, STAT1 induces a strong transcriptional response that is specific for GAS elements in target promoters. In contrast, transcription of many STAT3 target genes can also be regulated by several other STAT3-independent mechanisms. As for STAT3, we observe a positive correlation between phosphoprotein levels (phSTAT3/STAT3) and the signature activation. We observe only a weak correlation with STAT3 total levels, which is, once again, expected since only pSTAT3 levels are regulated across our panel of breast cancer cell lines.

The second analysis suggested is included in Supplementary Figure 10 of the revised manuscript. In TCGA patient data, STAT1 activation signature strongly correlates ($R = 0.89$) with STAT1 mRNA levels. The correlation for STAT3 is lower, but significant ($R=0.45$): As for the analysis at the protein level, unfortunately TCGA data is incomplete. We could only assess correlations with phospho-STAT3 levels (no data is available for STAT1, or for total STAT3 levels). Notably, we observe a marked correlation (Pearson $R=0.38$) between STAT3PY705 and the ssGSEA score, indicating that the ssGSEA score indeed captures STAT3 activity. As a control, we show below that the correlation disappears when we contrast STAT1 ssGSEA with phospho-STAT3.

4. It is not clear whether the authors developed their own implementation of ssGSEA or used existing software available by the Broad Institute group who developed the method or other software packages available (e.g. in

Bioconductor). They should clarify and provide appropriate reference if an existing software implementation was used.

We apologize for the omission, which is now corrected in the revised manuscript. We used the software developed by the Broad, indeed.

Reviewer #4:

1. **Fig. 1E : the bar scale of IHC should be shown**

This has been corrected in Figure 1.

2. **Fig. 1F-G: the raw data (instead of the fold changes) must be depicted.**

We have addressed this and the % of cells per specified subgroup out of total number of live cells analyzed is now reflected in the figures.

3. **Fig. 6B: does the vaccine efficacy depend on the MHC class I expression levels of the ip immunizing cells? results of the immunizing capacities of the STAT1- or STAT3 –CRISPR cell counterparts?**

The reviewer raises an excellent point. Comparison of surface MHC class I expression levels on each of the cell line used for IP immunization shows that MHC class I levels are comparable between MT/ShcA^{+/+} (864 line) and MT/Shc^{2F/2F} (5372 line) used for immunization, even though only the latter is exquisitely sensitive to tumor vaccination. Indeed, MT/Shc^{313F/313F} (6738 line) breast cancer cells basally upregulate surface MHC class I expression levels relative to MT/ShcA^{+/+} cells, which we believe may contribute, in part to the increased sensitivity of this cell line to immunization strategies. The data showing relative MHC class I expression levels across each cell line can be found in Figure 2f, g. Discussion of this point can be found on page 11, lines 22-24 and page 12, lines 1-3 of the current manuscript.

The reviewer raises a valid point regarding analysis of the immunizing capacities of the STAT1- or STAT3-deficient cell counterparts. This study would require testing 8 panels of cell lines (MT/ShcA^{+/+} - Control; MT/ShcA^{+/+} - STAT1-CRISPR; MT/ShcA^{+/+} - STAT3-CRISPR; MT/Shc^{2F/2F} – Control; MT/Shc^{2F/2F} – STAT1-CRISPR; MT/Shc^{313F/313F} – Control; MT/Shc^{313F/313F} – STAT1-CRISPR; MT/Shc^{313F/313F} – STAT3-CRISPR), each in two cohorts of mice (PBS versus immunized). This is a large study, which we anticipate will require 4-5 months to complete. The immunization of the animals requires one month prior to mammary fat pad injection. Beyond this, we followed the animals for a three-month period post mammary fat pad injection to ensure that we could classify them as tumor free. Finally, we would require additional time for histological assessment of the resulting mammary glands or tumors. Thus, although we agree that this would be a very interesting experiment, we respectfully suggest that it is beyond the scope of the current manuscript and hope that the reviewer agrees with us.

4. **Fig. 6E: comment on the percentages (and n absolute numbers) of human BC primary tumors falling into each categories in the TCGA data base ?**

This has been addressed in the results section. The added text now reads (page 13; lines 16-20): Interestingly, we could identify comparable numbers of primary breast tumors falling into one of four categories: STAT1^{Low} (1st quartile)/STAT3^{Low} (1st quartile),

n=110 or (9.1%); STAT1^{Low} (1st quartile)/STAT3^{High} (4th quartile), n=58 (4.8%); STAT1^{High} (4th quartile)/STAT3^{Low} (1st quartile), n=61 (5%); or STAT1^{High} (4th quartile)/STAT3^{High} (4th quartile), n=91 (7.5%) ssGSEA signatures. These data suggest that the STAT1/STAT3 ratio is dynamically regulated in individual human breast tumors and support the potential for this ratio in predicting intrinsic immune surveillance and the likely sensitivity of breast tumors to immunotherapies.

Again, we thank each of the reviewers for their valuable input. We feel that we have adequately addressed most of their criticisms and that this revised manuscript does provide the novelty and biological relevance required for publication in *Nature Communications*. We look forward to hearing from you soon.

Reviewers' comments:

Reviewer #2 (Remarks to the Author):

The manuscript by Ahn, Ursini-Siegel and co-workers on how ShcA, via STAT1 and STAT3 activation, control the immune response to tumors are of high interest to the community, and contains a substantial amount of new data.

It continues to be a very densely written manuscript, but the revision has improved the data presentation and discussion. Nevertheless, not all my previous comments were fully addressed, and a few more editing issues persist.

Comments:

1. For Figure S1, tumor onset for lines 864, 6738, and 6203 transplanted to wild types are noticeable longer (about twice as long; likely statistically significant) for transplants to wild type host in the experiments in panels a and b than in panels c and d. Is this possibly related to differences in the backgrounds of the CD8 null mouse line versus the IFN-gamma lines? It is noticeable that particularly the 313F cell lines have different tumor onset curves in the wild type hosts in the two experiments. I don't suggest that the authors perform additional backcrosses, but it is cheap and fast to have the backcrossed lines tested for FVB SNPs and if there are differences in backcrossing then this could lead to differences in immune responses to the pure FVB cell lines (if these are pure FVB?)

2. Similarly, tumors with stable disease for the Shc2F cell lines only seems to appear on the wild type mice for the IFN γ -/- line, but not when transplanted to wild types for the CD8-/- line. Again, this could be due to differences in the level of background crossing (which can be at variable percentage FVB after 8 generations), leading to some additional immune responses. Please at least acknowledge this in the Discussion.

3. In my comments of the original manuscript, I pointed out that the x- and y-axis scales on Figs. 4e, 4f and S1e, should be the same across all cell lines so data obtained using the many different cell lines and host mice can be more easily compared. This has not been corrected even though the rebuttal letter states so (previous comments #19 and #23). The same concern regarding using uniform scales on x- and y-axis applies to Figs. 1c, 1d, and S1f.

4. The images in Fig. S6b and the quantification doesn't fit: it looks like IHC images for ShcA and Shc2F in the IFN γ -/- background possibly were mixed up.

5. It is unclear whether "percentage of XX+ cells" is percentage of total cells, or of something else. This applies to e.g., GZMB+ cells on Fig. 1e, and CD8+ cells on Figs. 1f and 1g.

6. Is there a miscalculation of % CD8+ cells on fig.1f? 0.03% CD8+ cells for 2F tumors doesn't fit with 12% GZMB+ cells on Fig. 1e for the same tumors in wild type background. 0.03% is also very low compared to what normally is reported in the literature for tumors of the MMTV-PyMT GEMMs.

7. For supplemental Fig. S8, what is "PPC" (used on the figure)? Is it related to "percentage of positively-stained pixels" referred to in the figure legend?

8. I still find that the discrepancy between elevated CXCL9 protein and mRNA levels for the 313F cells lines in vitro versus elevated CXCL9 mRNA levels only for the tumors derived from the Shc2F cells (Fig. 2h) is insufficiently explained/discussed. Can the authors exclude that they mixed up the samples, could the CXCL9 in 2F tumors come from other cells in the tumors? I find it odd that both the 2F and the 313F cancer cells completely alters expression of this factor - in opposite directions - when going from in vitro to in vivo. Given that CXCL9 is a key chemokine for e.g. recruitment of cross-presenting dendritic cells, understanding this discrepancy is important.

9. Although it is correct as stated on p. 9, line 187-188 that STAT3 loss had no effect on STAT1 levels, it should be mentioned that it had a major effect on phosphoSTAT1 levels in the 313F line, complicating the relationship between ShcA and STAT signaling beyond what the authors currently state.

10. The authors refer to the abbreviation "DE" in the text (e.g., p. 13 line 283) but "DM" on the accompanying figure. This is confusing – do they mean two different things? If so, this should be explained it better.

11. The authors replotted Fig. 5A to use the same scale with and without IFN. This is very helpful and it would be preferable to also do this for Fig. 4d.

Reviewer #3 (Remarks to the Author):

All concerns were addressed by the authors.

Reviewer #4 (Remarks to the Author):

Excellent. High priority. No more comments.

Reviewers' comments:

Reviewer #2 (Remarks to the Author):

The manuscript by Ahn, Ursini-Siegel and co-workers on how ShcA, via STAT1 and STAT3 activation, control the immune response to tumors are of high interest to the community, and contains a substantial amount of new data.

It continues to be a very densely written manuscript, but the revision has improved the data presentation and discussion. Nevertheless, not all my previous comments were fully addressed, and a few more editing issues persist.

Comments:

1. For Figure S1, tumor onset for lines 864, 6738, and 6203 transplanted to wild types are noticeable longer (about twice as long; likely statistically significant) for transplants to wild type host in the experiments in panels a and b than in panels c and d. Is this possibly related to differences in the backgrounds of the CD8 null mouse line versus the IFN-gamma lines? It is noticeable that particularly the 313F cell lines have different tumor onset curves in the wild type hosts in the two experiments. I don't suggest that the authors perform additional backcrosses, but it is cheap and fast to have the backcrossed lines tested for FVB SNPs and if there are differences in backcrossing then this could lead to differences in immune responses to the pure FVB cell lines (if these are pure FVB?)

We can attest to the fact that the IFN $\gamma^{+/-}$ and CD8 $^{+/-}$ mice were backcrossed simultaneously with the same cohort of FVB females. Thus, it is highly unlikely that there will be different SNPs between both transgenic lines. Moreover, we used F8 as a starting point to initiate our studies based on previous literature from a highly respected group in the field, in which various immune-deficient strains of mice (*RAG-1 $^{+/-}$* , *JH $^{+/-}$* , *CD4 $^{+/-}$* , *CD8 $^{+/-}$* , and *IL4R α $^{+/-}$*) were backcrossed onto an FVB background for 5-15 generations (Cancer Cell. 2009 16(2):91-102). In this manuscript F7 was specifically chosen for CD8 $^{+/-}$ animals, which is consistent with our study.

Rather, our data suggests that the experimental differences observed in tumor onset for each of our cell lines within different experiments (CD8 $^{+/+}$ vs CD8 $^{-/-}$; Figure S1a **compared to** IFN $\gamma^{+/+}$ vs IFN $\gamma^{-/-}$; Figure S1c) is technical in nature. We ask the reviewer to keep in mind that these experiments were performed at different times and tumor onset is based on caliper measurement studies. Therefore, these types of changes are not unexpected. To further substantiate this argument, we ask the reviewer to compare the tumor onset curves for the independent cell lines in CD8 $^{+/+}$ vs CD8 $^{-/-}$ mice in Figure S1a and Figure 4e. In the latter, the timing of tumor onset for each cell line is much more in line with the kinetics observed with the IFN $\gamma^{+/+}$ vs IFN $\gamma^{-/-}$; study in question (Figure S1c).

2. Similarly, tumors with stable disease for the Shc2F cell lines only seems to appear on the wild type mice for the IFN $\gamma^{-/-}$ line, but not when transplanted to wild types for the CD8 $^{-/-}$ line. Again, this could be due to differences in the level of background crossing (which can be at variable percentage FVB after 8 generations), leading to some additional immune responses. Please at least acknowledge this in the Discussion.

By stable disease, we do not mean to imply that those tumors will not at some point re-acquire the ability to progressively grow. Instead, we observe that **a subset of Shc2F tumors** proceed through a plateau phase of tumor growth, which we believe to be immune mediated (all Shc2F tumors progressively grow in CD8^{-/-} and IFN γ ^{-/-} backgrounds). In fact, upon closer inspection, subsets of tumors (3/8) in the CD8^{+/+} cohort do proceed through a plateau phase between days 10-20 post palpation (see Figure 1c), which is not evident in control CD8^{-/-} animals. This represents 38% of the MT/Shc2F/2F tumors in CD8^{+/+} mice, which phenocopies closely this population in IFN γ ^{+/+} animals (40%). While the IFN γ ^{-/-} cohort was terminated earlier, the CD8 study was allowed to proceed until day 25 showing that these tumors do eventually acquire the ability to escape immune surveillance, which is not surprising, given that they surpassed 500mm³. We have added additional text to the results section (page 8 - blue font) to clarify this point.

3. In my comments of the original manuscript, I pointed out that the x- and y-axis scales on Figs. 4e, 4f and S1e, should be the same across all cell lines so data obtained using the many different cell lines and host mice can be more easily compared. This has not been corrected even though the rebuttal letter states so (previous comments #19 and #23). The same concern regarding using uniform scales on x- and y-axis applies to Figs. 1c, 1d, and S1f.

We apologize to the reviewer for this oversight. The x and y axis scales in Figures 1c, 1d, 4e, 4f, S1e and S1f have been corrected.

4. The images in Fig. S6b and the quantification doesn't fit: it looks like IHC images for ShcA and Shc2F in the IFN γ ^{-/-} background possibly were mixed up.

We have revised Fig. S6b to include more representative images.

5. It is unclear whether "percentage of XX+ cells" is percentage of total cells, or of something else. This applies to e.g., GZMB+ cells on Fig. 1e, and CD8+ cells on Figs. 1f and 1g.

The reviewer is correct that we are referring to percentage of total cells examined. This is clarified in the figure legends for Figure 1e, f, and g.

6. Is there a miscalculation of % CD8+ cells on fig.1f? 0.03% CD8+ cells for 2F tumors doesn't fit with 12% GZMB+ cells on Fig. 1e for the same tumors in wild type background. 0.03% is also very low compared to what normally is reported in the literature for tumors of the MMTV-PyMT GEMMs.

There are several factors that can explain the observed differences in % CD8+ cells by flow cytometry versus %GZMB+ cells when analyzed via immunohistochemistry. First, unlike other studies with MMTV/MT transgenic tumors (for example Cancer Cell. 2009 16(2):91-102), which first gated on CD45+ cells, we did not include the CD45 marker. Instead, the %CD8+ T cells are calculated based on the total number of cells analyzed. This was described in the supplementary materials and methods section of the manuscript. Second, one cannot compare the immune infiltrates in a mammary tumor directly isolated from

transgenic mice (which takes 3-4 months to reach clinical endpoint) to a tumor derived from a mammary fat pad injection, which reaches clinical endpoint within 3-4 weeks following first palpation. The differences in the kinetics of these growth curves alone will undoubtedly alter the immune-profiling of these tumors. Third, even following the Collagenase/Dispase treatment to disrupt epithelial interactions, many of the epithelial cells retain stable adherens junctions with their neighbours. Thus, during the filtration step, which removes these “epithelial clumps”, it is highly probable that infiltrating immune cells are also eliminated prior to commencement of flow cytometry. This is not an issue with immunohistochemistry. Thus, it may not be possible to directly compare ratios observed by flow cytometry versus immunohistochemistry. Finally, MT/Shc^{2F/2F} tumor growth is accelerated ~3 fold in CD8^{-/-} mice, relative to their wild-type counterparts. In contrast IFN γ -deficiency results in a 4 fold acceleration in tumor growth. This is observed with two independent MT/Shc^{2F/2F} cell lines (see Fig. 1 and Fig. S1). Thus, although CD8⁺ T cells are clearly essential for efficient immune surveillance in MT/Shc^{2F/2F} tumors, we cannot exclude the possibility that NK cells are not also activated (and therefore GZMB⁺) in these tumors. However, our genetic approach (using both CD8^{+/+} versus CD8^{-/-} mice) show that the CD8⁺ T cell compartment is the main driver of immune surveillance in MT/Shc^{2F/2F} tumors. This possibility is now included in the discussion (first paragraph, blue font).

7. For supplemental Fig. S8, what is “PPC” (used on the figure)? Is it related to “percentage of positively-stained pixels” referred to in the figure legend?

I believe the reviewer is referring to Figure S9. Indeed, PPC refers to the percentage of positively-stained pixels, which is now indicated in the supplementary figure legends.

8. I still find that the discrepancy between elevated CXCL9 protein and mRNA levels for the 313F cells lines in vitro versus elevated CXCL9 mRNA levels only for the tumors derived from the Shc2F cells (Fig. 2h) is insufficiently explained/discussed. Can the authors exclude that they mixed up the samples, could the CXCL9 in 2F tumors come from other cells in the tumors? I find it odd that both the 2F and the 313F cancer cells completely alters expression of this factor – in opposite directions - when going from in vitro to in vivo. Given that CXCL9 is a key chemokine for e.g. recruitment of cross-presenting dendritic cells, understanding this discrepancy is important.

We are certain that the observed phenotypes are not due to sample mix up. Rather, as described in the results section of the previous revision, this discrepancy was the precipitating force that lead us to examine whether ShcA signaling differentially regulated STAT1 versus STAT3 activation in the first place. In vitro, IFN-inducible CXCL9 expression in MT/Shc^{313F/313F} cells is tumor cell intrinsic but clearly cannot be sustained in vivo. Our subsequent in vivo data (Figures 3, 4 and Figure S6) show that heightened STAT3 signaling in MT/Shc^{313F/313F} tumors inhibits STAT1-driven anti-tumor immunity (and thus reduced CXCL9 expression in vivo). In contrast, MT/Shc^{2F/2F} cells do basally upregulate IFN-inducible CXCL9 expression in vitro but can be primed to induce this chemokine in mammary tumors in response to stromally-derived IFN γ signaling. Our subsequent in vivo data showing that MT/Shc^{2F/2F} tumors basally suppress STAT3 activation, which sensitizes them to stromally-derived IFN γ signaling in vivo.

We apologize if the description of the data was unclear and we have now added text to the results section (page 7, blue font) to clarify this point, which reads as follows:

“Combined, these data suggest that although reduced pY313-ShcA signaling potentiates IFN-regulated transcriptional responses, activation of an IFN-driven inflammatory response is insufficient to overcome immune suppression in MT/Shc^{313F/313F} tumors *in vivo* (Fig. 1c, d). In contrast, inhibition of pY239/240ShcA-coupled signaling pathways in mammary tumors (MT/Shc^{2F/2F}) specifically elicits immune surveillance *in vivo* (Fig. 1c, d; Supplementary Fig. 1), despite the fact IFN γ -driven signaling responses are basally reduced in these cell lines *in vitro* (Fig. 2). This is consistent with the fact that *IFN γ* and *CXCL9* mRNA levels are specifically increased in MT/Shc^{2F/2F} tumors but not in MT/ShcA^{+/+} or MT/Shc^{313F/313F} tumors (Fig. 2h). Thus, loss of pY313-ShcA signaling in breast cancer cells is sufficient to basally upregulate several IFN γ -inducible genes associated with anti-tumor immunity, including *CXCL9*, these IFN γ -inducible gene responses are restrained in MT/Shc^{313F/313F} mammary tumors *in vivo*. These data suggest that additional immunosuppressive mechanisms are engaged in MT/Shc^{313F/313F} tumors to restrain induction of IFN γ -driven anti-tumor immune responses. In contrast, MT/Shc^{2F/2F} tumors are sensitized to IFN γ -driven immune surveillance *in vitro* despite the fact that inhibition of pY239/240ShcA signaling cannot directly upregulate IFN γ -stimulated transcription *in vitro*. These data suggest that loss of pY239/240ShcA signaling downstream of tyrosine kinases must inhibit immunosuppressive signals in breast cancer cells and sensitize them to stromally-derived, IFN γ -inducible immune surveillance pathways *in vivo*.

To better understand these seemingly paradoxical observations and with the knowledge that IFN signaling requires the STAT1 transcription factor, we examined whether ShcA signaling dynamically regulates the activity of STAT1 and STAT3; two transcription factors with opposing roles in immune evasion”.

9. Although it is correct as stated on p. 9, line 187-188 that STAT3 loss had no effect on STAT1 levels, it should be mentioned that it had a major effect on phosphoSTAT1 levels in the 313F line, complicating the relationship between ShcA and STAT signaling beyond what the authors currently state.

As mentioned in the figure legend, the immunoblots shown in Figure 4b were performed in triplicate. The only reproducible and statistically significant difference observed over all three experiments was the increase in pY-705 STAT3 signaling observed following STAT1 loss, specifically in MT/ShcA cells. In contrast, deletion of STAT3 in MT/Shc313F cells showed increased STAT1 tyrosine phosphorylation in the absence of IFN γ in one of the three replicates with the other two showing no changes in the degree of Y701-STAT1 phosphorylation. This is evident from the quantification of the three experiments in Figure S7b. Moreover, STAT3 loss has no impact on total STAT1 levels in any of the triplicate experiments. Finally, pY701-STAT1 levels are comparable in all cell lines examined, irrespective of their STAT3 status. Thus, we do not believe that this difference in pY701-STAT1 levels is biologically relevant but included it in the immunoblot (original Figure 4b) for the sake of transparency. We apologize for the confusion that this may have caused and have now replaced the immunoblot panels Figure 4b with an independent experiment that more accurately reflects the data.

10. The authors refer to the abbreviation “DE” in the text (e.g., p. 13 line 283) but “DM” on the accompanying figure. This is confusing – do they mean two different things? If so, this should be explained it better.

This has been corrected in the text.

11. The authors replotted Fig. 5A to use the same scale with and without IFN. This is very helpful and it would be preferable to also do this for Fig. 4d.

The data in Figure 4d has been re-plotted as suggested by the reviewer.

**Reviewer #3 (Remarks to the Author):
All concerns were addressed by the authors.**

**Reviewer #4 (Remarks to the Author):
Excellent. High priority. No more comments.**

We feel that we have adequately addressed each of the criticisms raised by reviewer 2, both in the previous and current revision, and that this revised manuscript does provides the novelty, scientific rigor and biological relevance required for publication in *Nature Communications*.

REVIEWERS' COMMENTS:

Reviewer #2 (Remarks to the Author):

I have no further comments and look forward to see the study in print.